**METHOD**

# LinDA: linear models for differential abundance analysis of microbiome compositional data

Huijuan Zhou[1,2,3], Kejun He[3], Jun Chen[4*] and Xianyang Zhang[2*]

*Correspondence:
chen.jun2@mayo.edu;
zhangxiany@stat.tamu.edu
[4]Mayo Clinic, Rochester, USA
[2]Texas A&M University, College
Station 77843, USA
Full list of author information is
available at the end of the article

## Abstract

Differential abundance analysis is at the core of statistical analysis of microbiome data. The compositional nature of microbiome sequencing data makes false positive control challenging. Here, we show that the compositional effects can be addressed by a simple, yet highly flexible and scalable, approach. The proposed method, LinDA, only requires fitting linear regression models on the centered log-ratio transformed data, and correcting the bias due to compositional effects. We show that LinDA enjoys asymptotic FDR control and can be extended to mixed-effect models for correlated microbiome data. Using simulations and real examples, we demonstrate the effectiveness of LinDA.

**Keywords:** Compositional effect, Differential abundance analysis, False discovery rate, Multiple testing

## Background

The role of the human microbiome in health and disease has been intensively studied over the past few years, see, e.g., [1, 2], for several reviews. Potentially pathogenic or pro-biotic microorganisms can be identified by analyzing their abundances in a microbial ecosystem (e.g., the human gut) with respect to some covariate of interest such as disease status. Current prevailing technologies for studying the human microbiome use metagenomic sequencing, where either the DNA of a taxonomically informative gene (e.g., 16S rRNA) or all the genomic DNA in the microbial genome is sequenced. After obtaining the raw sequencing reads, the reads can be clustered into operational taxonomic units (OTUs), denoised into amplicon sequence variants (ASVs), or mapped to a microbial reference database (taxa) using existing bioinformatics pipelines such as UPARSE, DADA2, and MetaPhlAn [3–5]. For simplicity, we use the term taxon (pl. taxa) to represent any taxonomic unit (OTU/ASV/taxon) from a bioinformatics pipeline. Therefore, after bioinformatics processing, we have an abundance table recording the frequencies of detected

taxa in the samples, together with a meta data table capturing the sample-level information. Differential abundance analysis is then carried out based on the abundance and meta data table.

Ideally, we want to measure the absolute abundance of the microorganisms, i.e., the number of microorganisms per unit area/volume at the microbial ecosystem, and differential abundance analysis is performed on the absolute abundance data. However, the data from a sequencing experiment only captures the relative abundance (compositional) information since the total sequence read count, also known as sequencing depth or library size, does not reflect the total microbial load in the specimen due to the complex chemistry involved in sequencing [6, 7]. Although there are several experimental techniques such as qPCR, spike-in and flow cytometry that can be used to achieve absolute abundance measurement, the severe limitations of these techniques prevent their wide adoption [8]. Therefore, the prevailing sequencing protocol is still only able to measure the relative abundances. Drawing inferences about the changes on the unknown absolute abundance based on the measured relative abundance data is challenging due to missing the total microbial load information. The increase or decrease in the abundance of some taxa with respect to a covariate of interest automatically results in changes in the relative abundances of all other taxa, a statistical phenomenon known as compositional effects. Therefore, using the standard statistical techniques such as two-sample $t$-test, Wilcoxon rank sum test, and linear regression analysis ignoring the compositional nature of the data could lead to a large number of false discoveries.

For the differential abundance problem to be well defined, one has to make assumptions. One major assumption is that the differential signal is sparse, i.e., only a small proportion of taxa are associated with the covariate of interest. Although many studies have supported the sparse signal assumption, there are also studies support dense signal hypotheses, where a large number of taxa are differential with small effect sizes [9, 10]. Therefore, the validity of a method and the definition of true or false positive depends on the specific assumption one is willing to accept. Here our goal is to provide a statistical tool that could be potentially useful for pinpointing top candidate taxa for further biological validation.

To address compositional effects in differential analysis, one popular approach is robust normalization. It involves calculating a normalizing factor (scale factor), which is robust to a small number of differential taxa and could well capture the sequencing effort for the non-differential part. Therefore, dividing by such a normalizing factor will bring the abundance of the non-differential taxa to the same scale while retaining the differences for those differential ones. Assuming the number of differential taxa is small, different strategies have been used to calculate a robust normalizing factor including TMM, RLE, CSS, and GMPR [11–14]. We list these methods in Additional file 1: Table S1. In contrast, the naive total sum scaling (TSS) normalization, which divides the counts by the library size, is not a robust normalization method [14].

These normalization techniques can be combined with different statistical procedures in differential abundance analysis. For example, we can divide the counts by the normalizing factor from the normalization techniques in Additional file 1: Table S1 and then apply standard statistical tools based on the normalized data. The normalizing factor could also be included as an offset in regression models such as edgeR [15], DESeq2 [16], MicrobiomeDDA [17], and metagenomeSeq [13], where the TMM, RLE, GMPR, and CSS

normalization are the accompanying normalization methods. The recently developed MaAsLin2 [18] uses log linear models on the normalized abundance data. Different normalization approaches including TSS, TMM, CSS, and CLR are options in MaAsLin2. A variant to the robust normalization approach is to find a reference taxon or a set of reference taxa, which are assumed to be non-differential with respect to the covariate of interest. The data are then normalized by the count of the reference taxon (or the sum of the counts of the reference taxa). This strategy was used in RAIDA [19] and DACOMP [20].

Another line of methods to tackle the compositional effect uses (log) ratio approach since only ratios are well defined for compositional data [21]. The ALDEx2 method by [22] uses the centered log-ratio (CLR) transformation, where the counts of a sample are divided by their geometric mean before taking logarithms. Differential abundance analysis is then performed using Wilcoxon rank sum test or $t$-test based on the CLR transformed data. In the CLR approach, the geometric mean can also be regarded as a robust normalizing factor. The ANCOM proposed by [23] computes the pairwise ratios of the relative abundances and identifies the taxa with the most differential ratios. This is based on the observation that the abundance ratios for those differential taxa to other taxa are all differential assuming distinct effect sizes while the ratios for those non-differential taxa are mostly non-differential. Therefore, by analyzing the pattern of the pairwise ratios, one could distinguish the differential taxa from a background of non-differential taxa with high accuracy. Recently, a bias-corrected version of ANCOM (called ANCOM-BC) has been proposed [24], which uses a linear regression framework based on log-transformed taxa count and estimates the unknown bias term due to the compositional effect through an EM algorithm.

In the work of [25, 26], the authors evaluated several popular methods in differential abundance analysis (ANCOM-BC/MaAsLin2 not included) and showed that the inflation of the false discovery rate (FDR) is still a ubiquitous problem, and no method is satisfactory in all aspects. A method that is computationally efficient, relatively robust and powerful, and flexible enough to allow covariate adjustment and application to correlated microbiome data is still lacking in the field. In this paper, we propose a linear regression framework for differential abundance analysis (LinDA) to fill the methodological gap. LinDA involves three simple steps that can be carried out efficiently. First, it runs linear regressions using the CLR-transformed abundance data as the response. Then it identifies a bias term due to the compositional effect and corrects for the bias using the mode of the regression coefficients across different taxa. Finally, it computes the $p$ values based on the bias-corrected regression coefficients and applies the Benjamini-Hochberg (BH) procedure to control the FDR. We rigorously prove the asymptotic FDR control of the proposed method, making it the first procedure that enjoys a theoretical FDR control guarantee. Our approach is related to ANCOM-BC but differs in several aspects. (i) Our derivation provides a clear interpretation of the bias term and suggests a simple way to correct it. (ii) Our procedure does not involve the EM algorithm and can be 100–1000 times faster than ANCOM-BC in our numerical studies. (iii) Our method can be directly extended to the mixed-effect models. Longitudinal and repeated measurement-based microbiome studies have been increasingly common [27, 28] but statistical tools for correlated microbiome data analysis remain scarce. LinDA can analyze the correlated microbiome data using the classic linear mixed-effects models. Through extensive simulation studies and real data

analyses, we show that the new method outperforms the state-of-the-art approaches in terms of FDR control and power.

## Results
### Numerical studies
#### *Setups*
We conducted comprehensive simulations to evaluate the performance of the proposed method under different setups. We set $m = 500$ as the baseline for the number of taxa, which is similar to the number of tests at the species level for a typical microbiome study. We investigated the sample size $n = 50, 200$ representing small and large sample sizes, respectively. More combinations of $m$ and $n$ were studied in additional settings. We generated the absolute abundances from the log normal distribution and considered three cases for the covariate of interest and confounders: the covariate of interest follows the Bernoulli distribution and no confounder (denoted as C0), the covariate of interest follows the standard normal distribution and no confounder (C1), and the covariate of interest follows the Bernoulli distribution and two confounders (C2). In addition to the basic setting (log normal abundance distribution, denoted as S0), we investigated other settings to study the robustness of the proposed method including zero-inflated absolute abundances (S1), correlated absolute abundances (S2), gamma abundance distribution (S3), smaller $m$ (S4), smaller $n$ (S5), 10-fold difference in library size (S6), negative binomial abundance distribution (S7) and correlated microbiome data generated by mixed-effect model (S8). See the "Detailed setups for numerical studies" section for more details.

#### *Competing methods*
We compared our method with ANCOM-BC, ALDEx2, DESeq2, edgeR, metagenomeSeq and MaAsLin2. For DESeq2 and edgeR, we replaced their native normalization methods with GMPR normalization, which was shown to improve the power and false positive control in differential abundance analysis [14]. For metagenomeSeq, there are two implementations, `fitZig` and `fitFeatureModel`, in the R Bioconductor package `metagenomeSeq`. Currently, `fitFeatureModel` is only applicable to binary covariate case (C0). We use metagenomeSeq2 and metagenomeSeq to denote the `fitFeatureModel` and `fitZig` procedures, respectively. We also compared with the standard non-parametric methods: Wilcoxon rank sum test for case C0 (a binary covariate) and Spearman correlation test for case C1 (a continuous covariate), both with the GMPR normalized data.

For the proposed method, we considered two zero-handling approaches. The first approach adds a pseudo-count of 0.5 to all the counts, which is widely used in microbiome data analysis on the log scale. However, it has been shown to be problematic under certain situations [20]. We thus designed a new imputation-based approach, where the zeros were imputed by $N_s/(\max_{k:Y_{ik}=0} N_k)$ for $i$th taxon in $s$th sample, where $N_s$ denotes the library size (sequencing depth) of $s$th sample and $Y_{ik}$ denotes the read count of $i$th taxon in $k$th sample. In other words, zeros were imputed differently according to the library size of the sample, and zeros in the sample with a larger library size were replaced with larger fractions. In this imputation approach, we treat zeros as left-censored missing data. Suppose we only know the library sizes, then a natural strategy is to impute zeros in proportion to the library size with the sample of the largest library size receiving a fractional count close

to 1 (in our approach, we simply set it as 1). The purpose of the imputation strategy is to reduce false positives when the library size is correlated with the covariate of interest. As shown in the simulation studies, the pseudo-count approach worked sufficiently well in most settings except the setting S6, where the library size between the groups differed by 10-folds. In contrast, the imputation approach reduced the false positive rate extensively for the setting S6 (Additional file 2: Fig. S1). However, it was slightly less powerful than the pseudo-count approach when the library size was a not confounder (Additional file 2: Fig. S2). Thus, in the implementation, we used an adaptive approach: we first tested the association between the covariate of interest and the library size. If the *p* value was smaller than 0.1, we used the imputation approach conservatively; otherwise, we used the pseudo-count approach. Additional file 2: Fig. S1 and S2 show that the adaptive method controls the false positives when the library sizes are very different among groups while retaining the power when the library sizes are similar.

The proposed LinDA method can be viewed as a three-step procedure: CLR-OLS-BC (OLS stands for ordinary least squares and BC stands for bias correction), which can be easily extended to the linear mixed-effects model using CLR-LMM-BC (LMM stands for linear mixed-effect model). In the setting S8 (correlated microbiome data), we compared CLR-LMM-BC to CLR-OLS-BC, CLR-OLS, and CLR-LMM to demonstrate the utility of LinDA for correlated microbiome data analysis.

### Results

We use S0C0 (log normal abundance distribution, a binary covariate) to denote the setting S0 (log normal abundance distribution) with the covariate design C0 (a binary covariate) and likewise for other setups. For S0, we studied all the three covariate designs (C0–C2), and for S1–S8, we only performed C0 for demonstration. We found that DESeq2, edgeR and metagenomeSeq had severe FDR inflation under most settings. To increase the readability of the results (presented in figures), we did not include them in the main comparison and focused on the comparison between LinDA, ANCOM-BC, ALDEx2, metagenomeSeq2, MaAsLin2 and Wilcoxon (Figs. 1 and 2 and Additional file 2: Fig. S1–S15). Full results of all methods are presented in Additional file 2: Fig. S20–S30.

We first point out that MaAsLin2 with CLR normalization is essentially the same as the CLR-OLS procedure we described earlier. Additional file 2: Fig. S3 compares LinDA, CLR-OLS, MaAsLin2-TSS, MaAsLin2-TMM, MaAsLin2-CSS and MaAsLin2-CLR under the setting S0C0 (log normal abundance distribution, a binary covariate). We can see that MaAsLin2-CLR is close to CLR-OLS, both of which suffer from FDR inflation. We included MaAsLin2 with its default configuration (i.e., TSS normalization) in our comparisons below.

Figure 1 and Additional file 2: Fig. S4 and S5 show the results of the competing methods under the log normal abundance distribution with three covariate designs: a binary covariate (S0C0), a continuous covariate (S0C1), and a binary variable of interest with confounders (S0C2), respectively. Generally speaking, LinDA and ANCOM-BC have the best FDR and power trade-off. Under C0 and C2 (Fig. 1 and Additional file 2: Fig. S5), both methods control the FDR around the target level, and ANCOM-BC is slightly more powerful than LinDA when the sample size is small. However, under C1 (Additional file 2: Fig. S4), LinDA controls FDR at the target level at both sample sizes while ANCOM-BC

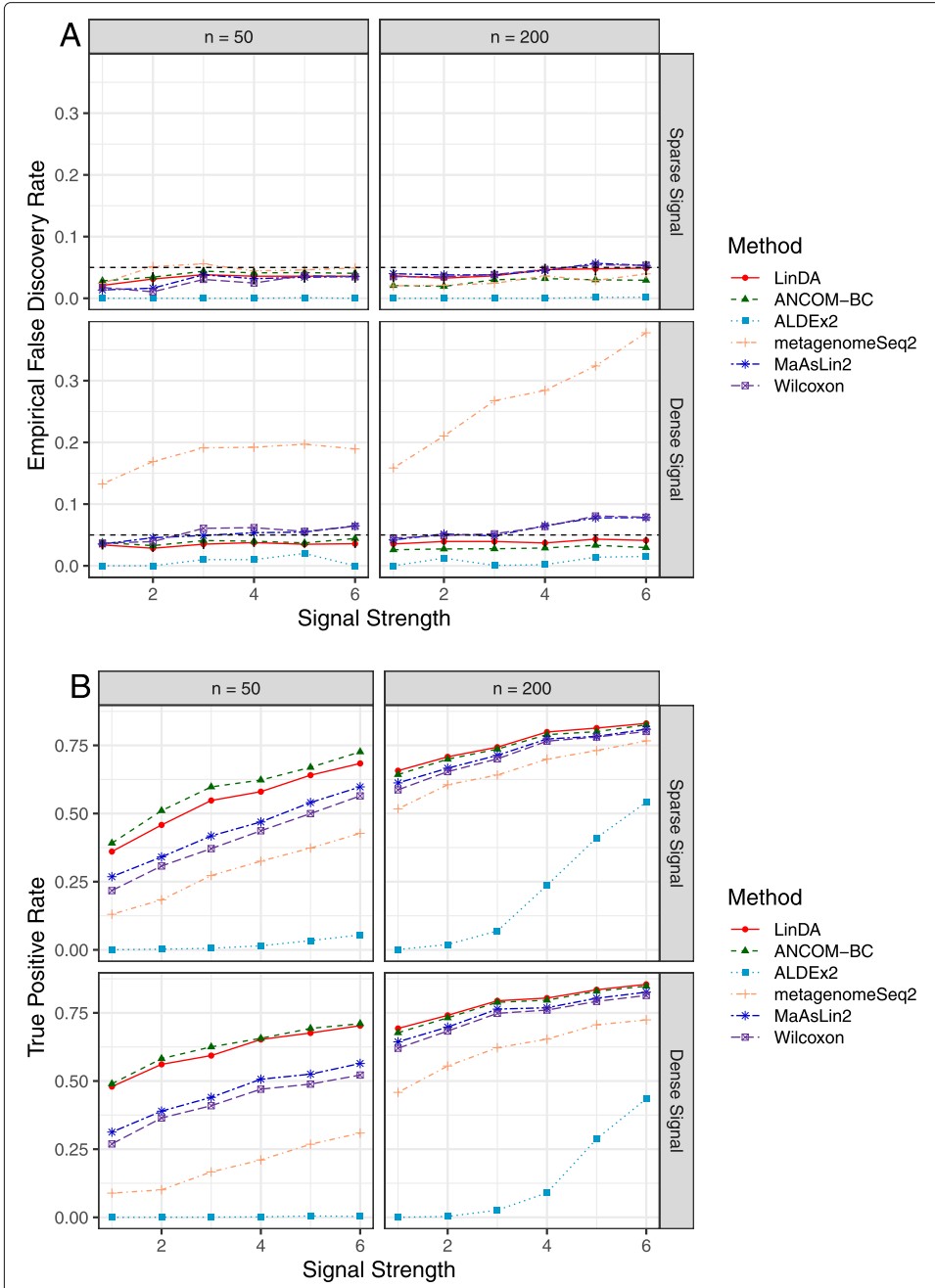

**Fig. 1** Performance comparison (S0C0: log normal abundance distribution, a binary covariate). Empirical false discovery rate (**A**) and true positive rates (**B**) were averaged over 100 simulation runs. Error bars (**A**) represent the 95% confidence intervals (CIs) of the method LinDA and the dashed horizontal line indicates the target FDR level of 0.05

has slight FDR inflation when the sample size is small. LinDA is also slightly more powerful than ANCOM-BC at a small sample size. The Wilcoxon rank sum test based on GMPR normalized data and MaAsLin2 perform well under C0 (a binary covariate, Fig. 1) with slightly inflated FDR at larger effect sizes and reasonable power across settings. In contrast, for a continuous covariate (C1, Additional file 2: Fig. S4), the Spearman rank correlation test and MaAsLin2 have large FDR inflation when the signal is dense. When there

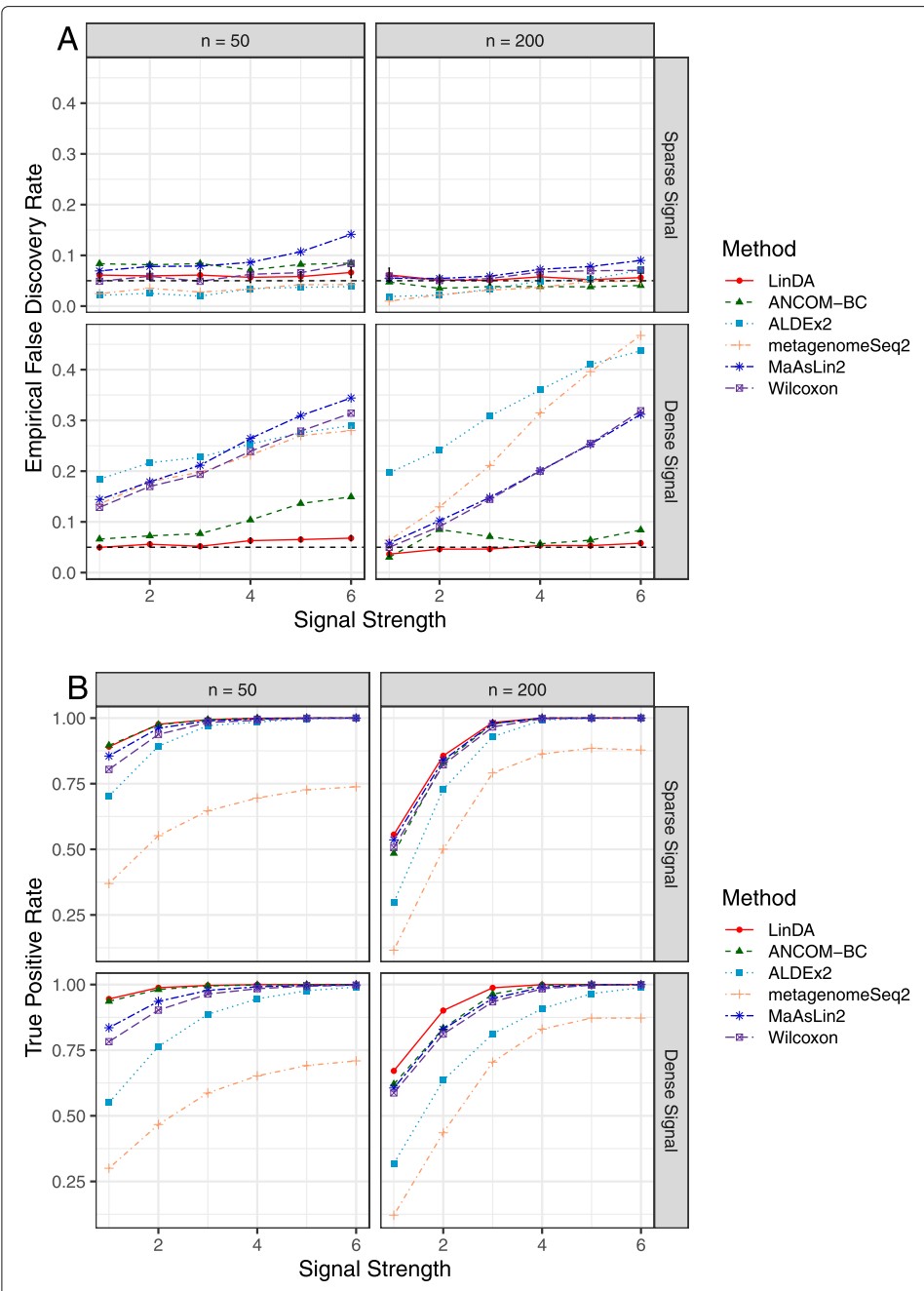

**Fig. 2** Performance comparison (S3C0: gamma abundance distribution, a binary covariate). Empirical false discovery rate (**A**) and true positive rates (**B**) were averaged over 100 simulation runs. Error bars (**A**) represent the 95% CIs of the method LinDA and the dashed horizontal line indicates the target FDR level of 0.05

are confounders (C2, Additional file 2: Fig. S5), Wilcoxon has severe FDR inflation when the sample size is large due to its inability to adjust for confounders, while MaAsLin2 provides acceptable results as under C0. ALDEx2 is a conservative method, which offers the strongest FDR control but is much less powerful. metagenomeSeq2 performs well when the signal is sparse but fails to control the FDR when the signal is dense. We also

studied the effect of zero inflation and the correlations among taxa (S1C0 and S2C0, Additional file 2: Fig. S6 and S7), where we observed similar patterns such that LinDA and ANCOM-BC had overall the best performance among the compared methods.

Since LinDA assumes a log normal distribution of the absolute abundance, it is interesting to evaluate its performance when the log normal assumption is violated. We thus simulated the absolute abundance data using a gamma distribution (S3C0), and the results are depicted in Fig. 2. It shows that LinDA controls the FDR close to the target level and has the highest power. When the signal is dense (20%), ANCOM-BC has a noticeable FDR inflation while ALDEx2, metagenomeSeq2, MaAsLin2 and Wilcoxon have severe FDR inflation when the signal is dense.

With a smaller number of taxa ($m = 50$, S4C0, Additional file 2: Fig. S8), ANCOM-BC controls the FDR and the power is also high. LinDA is the most powerful but it has slight FDR inflation. metagenomeSeq2, MaAsLin2, and Wilcoxon control the FDR but are less powerful in the case of sparse signal. However, when the signal is dense, they could not control the FDR properly. When the sample size is very small ($n = 20$ or 30, S5C0), LinDA controls the FDR around the target level and maintains high power (Additional file 2: Fig. S9). ANCOM-BC and metagenomeSeq2 have large FDR inflation and the inflation seems to increase as the sample size gets smaller. MaAsLin2 and Wilcoxon are much less powerful and ALDEx2 has virtually no power. Under the setting S6C0, where the sequencing depth differs by 10-folds, only ALDEx2 and our proposed method with adaptive zero-handling approach are able to control the FDR (Additional file 2: Fig. S10). LinDA achieves a better performance in both the FDR control and power than ALDEx2. It is interesting that ALDEx2 performs better under S6C0 than under other settings. We point out here that when we implemented ANCOM-BC, we disabled its zero treatment. To further investigate whether its zero treatment option improves its performance, we also run the procedure enabling its zero treatment (zero_cut = 0.9, lib_cut = 1000, struc_zero = TRUE), and found the results were very similar (S6C0, Additional file 2: Fig. S11).

Under the previous simulation settings, we found that DESeq2 and edgeR had the worst false positive control (Additional file 2: Fig. S20–S28). As the two methods assume negative binomial distribution for the counts, it is interesting to see their performance when the data are generated by their assumed model (S7C0). Additional file 2: Fig. S29 shows that DESeq2 and edgeR (and metagenomeSeq) remain to have serious FDR inflation, indicating that the normalization approach to address the compositional effect is not sufficient. In contrast, LinDA and ANCOM-BC perform the best among competitors as in other settings, and ANCOM-BC achieves higher power than LinDA (Additional file 2: Fig. S12).

Finally, we applied LinDA to correlated microbiome data (S8C0), where the other competing methods except MaAsLin2 are not applicable to correlated samples. Additional file 2: Fig. S13 and S14 compare the methods CLR-LMM-BC (LinDA-LMM), CLR-OLS-BC (LinDA-OLS), CLR-LMM, CLR-OLS, and MaAsLin2 for correlated data. In the scenario of comparing the pre-treatment and post-treatment samples (S8.1, Additional file 2: Fig. S13), we could clearly see that ignoring the bias tremendously increases the FDR level especially under dense signals (LinDA-LMM vs CLR-LMM). In addition, LinDA-LMM is more powerful than LinDA-OLS due to its ability to exploit the correlation between pre- and post-treatment samples. Under the replicate sampling setting (S8.2, Additional file 2: Fig. S14), we see that the LinDA-OLS has significant FDR inflation by

**Table 1** Runtime (in second) comparison under different settings (R version 4.0.3 (2020-10-10); Platform: x86_64-pc-linux-gnu (64-bit); CPU: E5-2670 v2 @ 2.50GHz; Memory: 67.7 GB). The result is based on one simulation run. The"elapsed" from the R command `system.time()` was used

|  |  | S0C0 | | S0C1 | | S0C2 | |
|---|---|---|---|---|---|---|---|
|  |  | LinDA | ANCOM-BC | LinDA | ANCOM-BC | LinDA | ANCOM-BC |
| $m = 500$ | $n = 200$ | **0.454** | 21.835 | **0.218** | 22.057 | **0.206** | 64.519 |
|  | $n = 10,000$ | **6.844** | 162.218 | **4.043** | 163.552 | **5.073** | 216.564 |
| $m = 5000$ | $n = 200$ | **1.598** | 184.972 | **1.607** | 162.611 | **1.615** | 599.985 |
|  | $n = 10,000$ | **28.253** | 5135.393 | **15.314** | 5157.148 | **15.494** | 5506.353 |

treating the replicate samples as independent ones. In contrast, LinDA-LMM controls the FDR at the target level. MaAsLin2 control the FDR under both settings but is less powerful than LinDA-LMM.

Based on the presented simulation settings, we summarize that LinDA and ANCOM-BC have overall the most robust performance among the methods evaluated. However, ANCOM-BC is computationally intensive. As shown in Table 1, LinDA could be 100–1000 times faster than ANCOM-BC, making LinDA a highly scalable method in practice. In addition, the extension of LinDA to the mixed-effect models is easily carried out and performs well.

### Real data applications

#### Datasets

We applied LinDA and the competing methods to three real datasets with independent samples from the studies of *C. difficile* infection (CDI, [29]), inflammatory bowel disease (IBD, [30]), and rheumatoid arthritis (RA, [31]). To demonstrate the use of LinDA on correlated microbiome samples, we applied LinDA to a dataset from the study of the smoking effect on the human upper respiratory tract (SMOKE, [32]). We used the microbiome samples from the throat for illustration, where each subject has two samples from the left and right sides of the throat. The CDI and RA datasets were provided by the authors while the IBD and the SMOKE datasets were downloaded from the Qiita database [33] with the study ID 1460 and 524. All the datasets have binary phenotypes. Antibiotics use is the confounder for the IBD dataset ($p = 0.03$ and OR = 0) while sex is the confounder for the SMOKE dataset ($p = 0.02$ and OR= 2.26). They will be adjusted in methods that are capable of covariate adjustment. We excluded samples with less than 1000 read counts and taxa which appear in less than 10% of the samples. The basic characteristics for the four filtered datasets are summarized in Table 2. We compared the detection power as well as their overlap patterns for LinDA, ANCOM-BC, ALDEx2, metagenomeSeq2, MaAsLin2, and Wilcoxon. Specifically, we compared the number of discoveries at different FDR levels

**Table 2** Characteristics of four real microbiome datasets. NORA represents new-onset untreated rheumatoid arthritis. The second and the third columns respectively list the number of taxa and sample size of each filtered dataset (prevalence $\geq$10%, library size $\geq$1000)

|  | *m* | *n* | *u* | *c* |
|---|---|---|---|---|
| CDI | 123 | 183 | CDI/Diarrhea control (94 v.s. 89) |  |
| IBD | 579 | 81 | Crohn's disease/Healthy (62 v.s. 19) | Antibiotic use (n/y, 48 + 19 v.s. 14 + 0) |
| RA | 438 | 72 | NORA/Healthy (44 v.s. 28) |  |
| SMOKE | 209 | 132 | Smoke (n/y, 67 v.s. 65) | Female/Male (31 + 16 v.s. 36 + 49) |

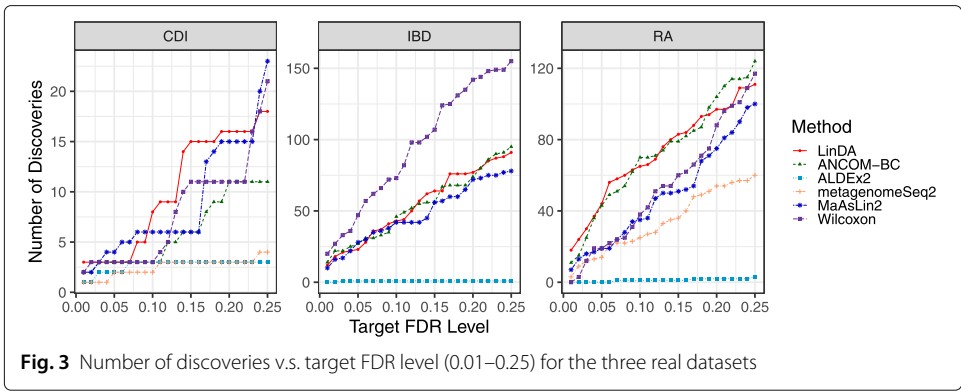

**Fig. 3** Number of discoveries v.s. target FDR level (0.01–0.25) for the three real datasets

(0.01–0.25) and used UpSet plot [34] to show the overlap at the target FDR of 0.1. We used winsorization at quantile 0.97 to reduce the impact of potential outliers as recommended in [17].

### Results

For the CDI dataset, LinDA or MaAsLin2 made the most discoveries at different FDR levels (Fig. 3). At 10% FDR, LinDA discovered eight and MaAsLin2 discovered six taxa associated with CDI. In contrast, ANCOM-BC, ALDEx2, and Wilcoxon discovered three while metagenomeSeq2 discovered two. As discussed in [29], subjects with CDI were more likely to have the bacterial family Lachnospiraceae and Erysipelotrichaceae. LinDA found one more taxon belonging to Lachnospiraceae than other methods (blue bars in Fig. 4). Besides, LinDA, MaAsLin2, and Wilcoxon found one differential taxon belonging

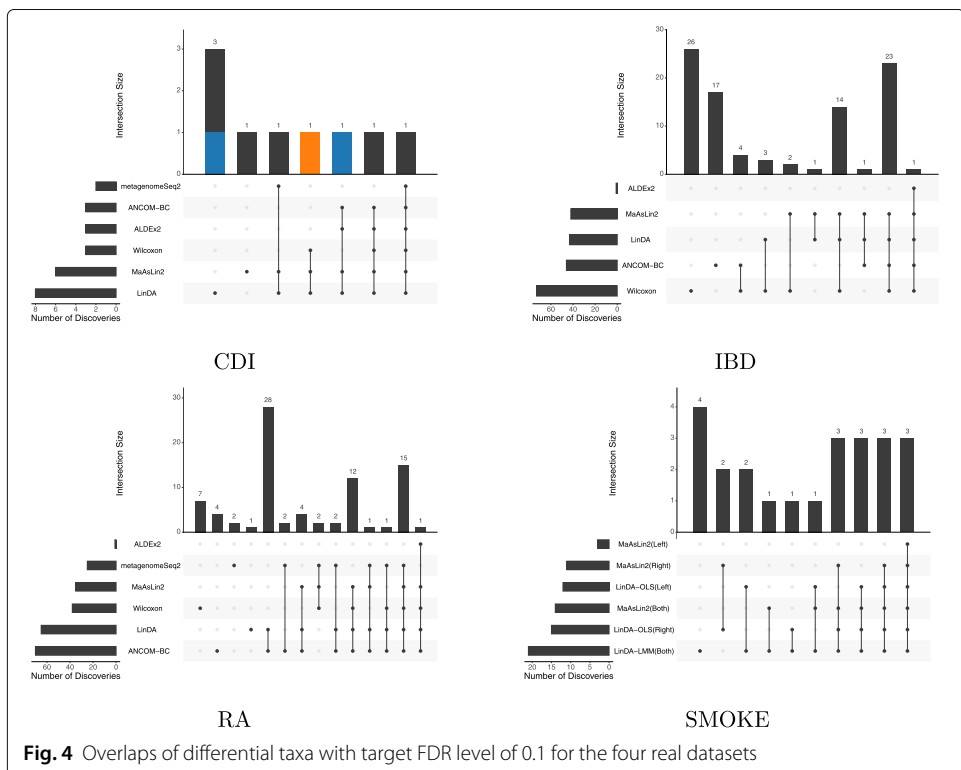

**Fig. 4** Overlaps of differential taxa with target FDR level of 0.1 for the four real datasets

to Erysipelotrichaceae while the other three methods did not identify any (orange bar in Fig. 4). For the IBD dataset, LinDA detected a similar number of taxa as ANCOM-BC and MaAsLin2. Wilcoxon rank sum test detected a large number of taxa associated with the disease status, but this could be due to the confounding effects of antibiotics use since it could not adjust for covariates. From Fig. 4, we observe that most discoveries by LinDA are shared by ANCOM-BC, MaAsLin2, or Wilcoxon. For the RA dataset, LinDA detected a similar number of taxa as ANCOM-BC and more taxa than other methods. The differential taxa detected by LinDA and ANCOM-BC largely overlapped. Overall, the results are consistent with the simulation studies.

Finally, we applied LinDA-LMM and MaAsLin2 to the SMOKE dataset, where each subject has two replicate samples from the throat. The aim is to identify smoking-associated taxa adjusting for the sex. To account for the correlation between the replicate samples, we included a subject-level random intercept in LinDA-LMM. As a comparison, we also applied LinDA-OLS and MaAsLin2 to the right and left throat samples separately. LinDA-OLS based on the left or right throat samples alone discovered 12 and 15 differential taxa at 10% FDR. When both left and right samples were used in LinDA-LMM, 21 differential taxa were identified, covering the majority of the taxa identified based on the left or right throat samples alone (Fig. 4). In addition, LinDA-LMM detected five taxa that were missed by analyzing the left or right samples separately. Compared to MaAsLin2, LinDA-LMM discovered seven more differential taxa. Therefore, LinDA-LMM provides a convenient way to analyze correlated microbiome datasets and enjoys the power improvement by analyzing all samples together.

To visualize the results, `LinDA` provides a function to generate the effect size plot and volcano plot for differential taxa. Additional file 2: Fig. S16–S19 display the effect size plots of differential taxa at FDR level of 0.1 and volcano plots for the four datasets, respectively. In the effect size plots (Additional file 2: Fig. S16A–S19A), the taxa in black are detected by LinDA and taxa in red are detected solely by LinDA. In Additional file 2: Fig. S16A (CDI), the taxa in blue are missed by LinDA but detected by one or more other methods. In Additional file 2: Fig. S17A (IBD) and S18A (RA), the taxa in blue are missed by LinDA but detected by two or more other methods. No taxa are detected by MaAsLin2 but missed by LinDA-LMM for the SMOKE dataset. Based on the effect size plots for the CDI, IBD and RA datasets, we can see that, for the taxa solely detected by LinDA, the effect sizes tend to be underestimated without bias correction and bias correction improves the power in these cases. On the contrary, for the taxa missed by LinDA, the effect sizes tend to be overestimated without bias correction. In addition, we observe that the differential taxa for the IBD, CDI, and RA datasets are more unbalanced (i.e., more negative or positive associations) while the differential taxa for the SMOKE dataset are relatively balanced (i.e., similar numbers of negative and positive associations). Indeed, the effect size plots, where we plot both the debiased and un-debiased coefficients, revealed larger biases for the IBD, CDI, and RA datasets.

## Discussion

Differential abundance analysis is at the core of the statistical analysis of microbiome data. Microbiome data are compositional in nature and all we know are the relative abundances, making the identification of differentially abundant taxa at the ecological site particularly challenging [6, 7]. Numerous differential abundance analysis methods have

been proposed focusing on addressing the compositional effects [13, 15–17, 19, 20, 22–24]. Among all the competing methods, ANCOM-BC is the state-of-the-art method, it has been demonstrated to be more robust and powerful than the competing methods. However, there are two drawbacks of ANCOM-BC. First, it is computationally intensive for large-scale microbiome datasets such as the AmericanGut dataset. Due to the huge inter-subject variation, large-scale microbiome studies have been increasingly popular, resulting in larger sample sizes. On the other hand, metagenomic sequencing has become increasingly deeper to have a high-resolution view of the microbiome, leading to an unprecedented number of new microbial features. To meet the analysis need for such large-scale datasets, a computationally efficient tool is much needed. Secondly, ANCOM-BC is not applicable to correlated/clustered microbiome datasets such as those from family/longitudinal microbiome studies or studies with paired and repeated measurements [27, 28]. Longitudinal microbiome analysis, which enables the study of the trajectory of the microbiome as well as controls for potential confounders, has been increasingly employed in human microbiome studies. Unfortunately, statistical tools for longitudinal microbiome studies are scarce. In contrast, LinDA is computationally efficient since it only involves fitting regular linear regression models and could be easily scaled to hundreds of thousands of taxa. Moreover, the extension of LinDA to linear mixed-effects models (LMM) is straightforward and we have highly efficient tools such as the R `lme4` package [35] for fitting LMM. Therefore, differential abundance analysis of correlated/clustered microbiome datasets could be easily performed using LinDA. Our framework also gives more insights into the CLR-based approach, which has been widely used in compositional data analysis [21]. However, the bias of CLR regression models has not been formally recognized to our best knowledge. Our framework justifies the use of CLR regression and provides a solution to correct the bias associated with CLR regression.

In the simulation, we found that Wilcoxon rank sum test and MaAsLin2 showed similar FDR/power curves and performed fairly well in most settings. As we mentioned earlier, MaAsLin2 is based on log linear models on the normalized count data, and it is essentially a two-sample $t$-test when no confounders are included, which explains why MaAsLin2 is close to Wilcoxon rank sum test. However, when we simulated an even stronger compositional effect by drawing the differential taxa from the top 25% most abundant taxa, we found that Wilcoxon rank sum test and MaAsLin2 began to break down (Additional file 2: Fig. S15). ANCOM-BC was overall robust and powerful, but it had inflated type I error at small sample sizes. metagenomeSeq2 did not perform well when the signal was dense and was generally less powerful than ANCOM-BC and LinDA. ALDEx2 was the most conservative method: its strong FDR control was at the expense of statistical power. LinDA was as competitive as ANCOM-BC in most settings. It had better FDR control than ANCOM-BC when the sample size was small or the covariate of interest was continuous. However, LinDA had some FDR inflation when the number of taxa was small. Under a very strong compositional effect (Additional file 2: Fig. S15), LinDA also showed some FDR inflation but overall it had the best FDR and power trade-off.

When the library size was associated with the covariate of interest, all existing methods had severe type I error inflation. Fortunately, such association is detectable and if we see a significant association, rarefaction should be used for those methods. Although rarefaction controls the effect of uneven library sizes, it discards a significant portion of

the reads and thus loses much information in the data. When there are many samples with small library sizes, the users have to decide whether to retain more reads or more samples. In LinDA, we implemented a heuristic imputation method, where the imputed values are proportional to the library sizes. This procedure makes the imputed data after CLR transformation independent of the library size and substantially reduces the inflated type I error due to library size confounding.

Although the presented simulation settings could give basic insights into the performance of various methods, such model-based simulations might not be able to capture the full characteristics of the real microbiome data. It is very likely that the performance of the compared methods will change using a different simulation framework. Moreover, our simulation strategy purposely creates strong compositional effects, where all differential taxa show the same direction of change. Such setting is used to test the limit of the various methods in addressing the compositional effects. However, in real data, the compositional effects may not be always strong, and the FDR inflation of many methods could be very moderate. Therefore, a future benchmarking study, which uses real data-based simulation strategy and investigates all biologically plausible differential settings, is much needed to have a comprehensive and objective evaluation of existing differential abundance analysis methods.

As for all model-based approaches, LinDA has several assumptions and limitations. First, LinDA relies on the assumption that there is a mode at 0 for the regression coefficients (Condition (vi) in Theorem 1). This assumption is easy to be met if the signal is sparse. In the simulation, we show that when the signal density is around 20%, LinDA is still very robust. However, when the signal is extremely dense, LinDA could fail. Second, LinDA assumes a log linear model on the absolute abundance. Although this is a reasonable assumption, which has been widely adopted in the analysis of abundance data, the interaction between the host and the microbiome could be more complex than the simple log linear relationship. Analysis of the residuals from the CLR regression could provide evidence about whether the assumption is reasonable. If the model assumption is violated, a permutation test or transformation of the variables may be performed. Finally, although LinDA provides asymptotic FDR control, its finite-sample FDR control is not guaranteed. Based on numerical simulations, we found that LinDA performed well under small sample sizes. However, we did observe some FDR inflation under a small feature size due to inefficiency in mode estimation with few features. Therefore, we do not recommend applying LinDA to datasets with small feature sizes (e.g., $m < 50$) such as phylum-level abundance data.

LinDA uses the relative abundance data and does not model the sampling variability of the read counts. This could reduce the statistical power for those less abundant taxa, whose sampling variability is larger than those abundant taxa. To remedy the power loss, another multinomial sampling layer could be imposed on top of LinDA. However, the computational complexity will be increased significantly, breaking the simplicity of LinDA. Another approach is to perform posterior inference of the underlying proportions based on a Bayes approach. Once we obtain the posterior samples, LinDA can be applied to the posterior samples and results are then aggregated, similar in the spirit to the multiple imputation method [36].

Besides microbiome data, LinDA could be applied to other sequencing data such as RNA-Seq data since all sequencing data are compositional in nature [37]. Thus, LinDA

could be an alternative for differential expression analysis if there are strong compositional effects, for example, when the highly abundant genes are differential with the same direction of change.

Finally, we comment that addressing compositionality is more relevant when analyzing individual microbial features such as differential abundance analysis, since the major interest to biomedical investigators is to find those truly differential features ("driver") instead of those driven by the compositional effect ("passenger"). However, for community-wide analysis such as distance-based analysis [38, 39], addressing the compositionality may not be necessary in order to control the type I error. This is because that compositional effect is only relevant under the alternative hypotheses. Considering compositionality in the community-wide analysis has also been found to have small effects on the statistical power [25, 40]. Additionally, in microbiome-based predictive models [41], the relative abundances and/or their ratios could already be informative features for prediction and addressing compositionality may not necessarily increase the prediction accuracy significantly. Therefore, whether to address compositionality depends on the specific problems.

## Conclusions

In summary, we proposed LinDA for differential abundance analysis of microbiome compositional data. LinDA identified a bias associated with traditional linear regression models based on CLR-transformed abundance data and proposed a strategy to estimate and correct the bias. LinDA can be extended to linear mixed-effects model for analysis of correlated microbiome data. As a general methodology, LinDA can be applied to differential abundance analysis of other high-dimensional compositional data.

## Methods

### Setup

We use $C$, $C_1$, and $C_2$ to denote positive constants, which can be different from line to line. As summarized in the background, there are two ways to tackle the compositional effects in differential abundance analysis, namely normalization and log-ratio transformation. In this paper, we adopt the CLR transformation and develop a bias-correction procedure to address the compositional effects. Denote the absolute abundance and the observed read count of the $i$th taxon in the $s$th sample by $X_{is}$ and $Y_{is}$, respectively. For the $s$th sample, the total read count of all taxa, $N_s = \sum_{i=1}^{m} Y_{is}$, is determined by the sequencing depth and DNA materials. Given $N_s$, it is natural to model the stratified count data over $m$ taxa through a multinomial distribution as

$$P(Y_{1s} = y_{1s}, \ldots, Y_{ms} = y_{ms}) = \frac{N_s!}{\prod_{i=1}^{m} y_{is}!} \prod_{j=1}^{m} \left( \frac{X_{js}}{\sum_{i=1}^{m} X_{is}} \right)^{y_{js}} \tag{1}$$

Under (1), we have

$$\log \left( \frac{Y_{is}}{\sum_{j=1}^{m} Y_{js}} \right) = \log \left( \frac{X_{is}}{\sum_{j=1}^{m} X_{js}} \right) + e_{is}, \tag{2}$$

where $e_{is}$ denotes the estimation error, which is expected to diminish as $N_s$ gets large.

## OLS estimation

We consider the log linear model on the absolute abundance

$$\log(X_{is}) = u_s \alpha_i + (1, \mathbf{c}_s^\top)\boldsymbol{\beta}_i + \epsilon_{is}, \tag{3}$$

where $\mathbf{c}_s = (c_{s1}, ..., c_{sd})^\top$ is the $d$-dimensional covariates to be adjusted, $u_s$ is the covariate of interest, and $\epsilon_{is}$ is the error term. Our goal is to discover taxa that are differentially abundant with respect to $u_s$. Statistically, we want to simultaneously test the following $m$ hypotheses

$$H_{0,i} : \alpha_i = 0 \text{ vs. } H_{a,i} : \alpha_i \neq 0.$$

Set $\varepsilon_{is} = \epsilon_{is} + e_{is}$. Under (2) and (3), the CLR-transformed data satisfies the following linear model

$$
\begin{aligned}
W_{is} := \log\left\{\frac{Y_{is}}{(\prod_{j=1}^m Y_{js})^{1/m}}\right\} &= \log\left(\frac{Y_{is}}{\sum_{k=1}^m Y_{ks}}\right) - \frac{1}{m}\sum_{j=1}^m \log\left(\frac{Y_{js}}{\sum_{k=1}^m Y_{ks}}\right) \\
&= \log(X_{is}) - \frac{1}{m}\sum_{j=1}^m \log(X_{js}) + e_{is} - \frac{1}{m}\sum_{j=1}^m e_{js} \\
&= u_s(\alpha_i - \bar{\alpha}) + (1, \mathbf{c}_s^\top)\left(\boldsymbol{\beta}_i - \bar{\boldsymbol{\beta}}\right) + \varepsilon_{is} - \bar{\varepsilon}_s,
\end{aligned}
\tag{4}
$$

where $\bar{\alpha} = m^{-1}\sum_{i=1}^m \alpha_i$, $\bar{\boldsymbol{\beta}} = m^{-1}\sum_{i=1}^m \boldsymbol{\beta}_i$, and $\bar{\varepsilon}_s = m^{-1}\sum_{i=1}^m \varepsilon_{is}$. From (4), we can see that the OLS estimator for $\alpha$ based on the CLR-transformed data is biased with the bias term being $\bar{\alpha}$. Let $\bar{\alpha}_i = \alpha_i - \bar{\alpha}$, $\bar{\boldsymbol{\beta}}_i = \boldsymbol{\beta}_i - \bar{\boldsymbol{\beta}}$, $\bar{\varepsilon}_{is} = \varepsilon_{is} - \bar{\varepsilon}_s$, and $\bar{\sigma}_i^2 = \mathrm{var}(\bar{\varepsilon}_{is})$. Denote by $\tilde{\alpha}_i$, $\tilde{\boldsymbol{\beta}}_i$, and $\hat{\sigma}_i^2$ the OLS estimators of $\bar{\alpha}_i$, $\bar{\boldsymbol{\beta}}_i$, and $\bar{\sigma}_i^2$, respectively. We then have

$$(\tilde{\alpha}_i, \tilde{\boldsymbol{\beta}}_i^\top)^\top = \left(\sum_{s=1}^n \mathbf{z}_s \mathbf{z}_s^\top\right)^{-1}\left(\sum_{s=1}^n \mathbf{z}_s W_{is}\right), \quad \hat{\sigma}_i^2 = \frac{1}{n-d-2}\sum_{s=1}^n \left\{W_{is} - \left(\tilde{\alpha}_i, \tilde{\boldsymbol{\beta}}_i^\top\right)\mathbf{z}_s\right\}^2, \tag{5}$$

where $\mathbf{z}_s = (u_s, 1, \mathbf{c}_s^\top)^\top$. We respectively let $\mathrm{var}_{\mathbf{z}}(\cdot)$ and $\mathrm{cov}_{\mathbf{z}}(\cdot, \cdot)$ denote the variance and covariance computed conditional on $\mathbf{z}_1, ..., \mathbf{z}_n$. It can be shown that

$$\mathrm{var}_{\mathbf{z}}(\tilde{\alpha}_i) = \hat{\rho}n^{-1}\bar{\sigma}_i^2 = \hat{\rho}n^{-1}m^{-1}\left\{(m-2)\sigma_i^2 + m^{-1}\sum_{i=1}^m \sigma_i^2\right\},$$

$$\mathrm{cov}_{\mathbf{z}}(\tilde{\alpha}_i, \tilde{\alpha}_j) = \hat{\rho}n^{-1}m^{-1}\left\{-(\sigma_i^2 + \sigma_j^2) + m^{-1}\sum_{i=1}^m \sigma_i^2\right\}, \quad \text{for } i \neq j,$$

where $\hat{\rho}$ is the $(1,1)$th element of $(n^{-1}\sum_{s=1}^n \mathbf{z}_s \mathbf{z}_s^\top)^{-1}$.

## Bias correction

In many applications, it is reasonable to assume that there is only a small portion of differential taxa, i.e., most $\alpha_i$'s are equal to 0. Under this assumption, as $\tilde{\alpha}_i$ is an unbiased estimator for $\bar{\alpha}_i = \alpha_i - \bar{\alpha}$, the mode of $\tilde{\alpha}_i$ is expected to be close to $-\bar{\alpha}$. This observation motivates us to estimate $-\bar{\alpha}$ by

$$-\tilde{\alpha} = \frac{\widehat{\mathrm{mode}}(\{\sqrt{n}\tilde{\alpha}_i\}_{i=1}^m)}{\sqrt{n}}, \quad \text{where } \widehat{\mathrm{mode}}(\{\sqrt{n}\tilde{\alpha}_i\}_{i=1}^m) = \underset{x\in\mathbb{R}}{\mathrm{argmax}}\frac{1}{mh}\sum_{i=1}^m K\left(\frac{x - \sqrt{n}\tilde{\alpha}_i}{h}\right). \tag{6}$$

In (6), $K$ is a non-negative even function with $\int_{-\infty}^{\infty} K(y)dy = 1$, and $h$ is the bandwidth parameter. Under some regular conditions, we have

$$\sqrt{n}(\tilde{\alpha} - \bar{\alpha}) = o_{\mathbb{P}}(1)$$

as $m, n \to \infty$ (see the supplementary material for the proof). Therefore, one can estimate $\alpha_i$ by the bias-corrected estimator $\hat{\alpha}_i = \tilde{\alpha}_i + \tilde{\alpha}$.

**Testing procedure**

To construct a statistic for testing $H_{0,i}$, we need to find a proper estimator for the variance of $\hat{\alpha}_i$. To this end, we note that

$$\text{var}_{\mathbf{z}}(\hat{\alpha}_i) = \text{var}_{\mathbf{z}}(\tilde{\alpha}_i) + \text{var}_{\mathbf{z}}(\tilde{\alpha}) + 2\text{cov}_{\mathbf{z}}(\tilde{\alpha}_i, \tilde{\alpha}).$$

Since $\text{var}_{\mathbf{z}}(\tilde{\alpha}_i)$ is $\hat{\rho}\hat{\sigma}_i^2/n$, it dominates $\text{var}_{\mathbf{z}}(\tilde{\alpha})$ and $\text{cov}_{\mathbf{z}}(\tilde{\alpha}_i, \tilde{\alpha})$ as $n, m \to \infty$ under mild conditions. Thus, we estimate the variance of $\hat{\alpha}_i$ by $\hat{\rho}\hat{\sigma}_i^2/n$. As shown in the next section, the studentized statistic $T_i := \sqrt{n}\hat{\alpha}_i/\sqrt{\hat{\rho}\hat{\sigma}_i^2}$ is asymptotically normal. However, for small sample, we found that $t$-distribution provides a better approximation to the sampling distribution of $T_i$. We define the $p$ value for testing $H_{0,i}$ as

$$p_i = 2F_{n-d-2}(-|T_i|), \tag{7}$$

where $F_{n-d-2}(\cdot)$ denotes the cumulative distribution function of $t$-distribution with $n - d - 2$ degrees of freedom. Based on the $p$ values in (7), we can use the BH procedure to control the FDR. The above discussion leads to the following Algorithm 1.

---

**Algorithm 1** Linear models for differential abundance analysis (LinDA)

1.  Step 1: Run OLS based on the CLR transformed observations and calculate $\tilde{\alpha}_i$ and $\hat{\sigma}_i^2$ as in (5).
2.  Step 2: Compute the bias-corrected estimates $\hat{\alpha}_i = \tilde{\alpha}_i + \tilde{\alpha}$ with $\tilde{\alpha}$ defined in (6).
3.  Step 3: Calculate the $p$ values as in (7) and run the BH procedure.

---

**Remark 1** *Built upon the linear regression framework, our method could be easily extended to the mixed-effect model:*

$$\log(X_{is}) = u_s\alpha_i + (1, \mathbf{c}_s^\top)\boldsymbol{\beta}_i + \mathbf{r}_s^\top\boldsymbol{\gamma}_i + \varepsilon_{is},$$

*where $\boldsymbol{\gamma}_i$ is the random effect and $\mathbf{r}_s$ is the corresponding design. Mixed-effects can be used to analyze correlated microbiome data from studies involving replicates or spatial sampling as well as family-based and longitudinal microbiome studies. We suggest using the R function* `lmer` *to estimate the parameters for the CLR-transformed data. Denote by $\tilde{\alpha}_{i,lmer}$, $\hat{\sigma}_{i,lmer}^2$, and $df_{i,lmer}$ the estimations for $\bar{\alpha}_i$, the variance of $\tilde{\alpha}_{i,lmer}$, and the degrees of freedom of $\tilde{\alpha}_{i,lmer}$ from the* `lmer` *function. We compute the bias-corrected estimates $\hat{\alpha}_{i,lmer} = \tilde{\alpha}_{i,lmer} + \tilde{\alpha}_{lmer}$, where $\tilde{\alpha}_{lmer}$ is obtained as the same procedure used in (6). Similarly, we let $T_{i,lmer} = \hat{\alpha}_{i,lmer}/\hat{\sigma}_{i,lmer}$ and $p_{i,lmer} = 2F_{df_{i,lmer}}(-|T_{i,lmer}|)$. The BH procedure on $p_{i,lmer}$ is finally used to control the FDR.*

**Remark 2** *Compared to the existing methods based on either normalization or CLR transformation, our method is computationally much more efficient and can be easily*

*scaled to problems with tens of thousands of taxa. Table 1 compares the computation time of LinDA and ANCOM-BC based on simulated datasets. We observe that our method is 100–1000 times faster than ANCOM-BC. We also tested on a massive dataset of the similar scale of the AmericanGut project [42] ($m = 5000$ and $n = 10,000$). ANCOM-BC completed the analysis in 85 min compared to 28 s for our method (see the column of S0C0 in Table 1). Large-scale microbiome studies have been increasingly common to overcome the large inter-subject variability, making our method practically useful for the analysis of big microbiome datasets.*

### Asymptotic FDR control

Suppose the target FDR controlling level is $q$. The BH procedure is equivalent to finding the smallest $t^*$ such that $\widehat{\text{FDP}}(t^*) \leq q$, where

$$\widehat{\text{FDP}}(t) = \frac{2m F_{n-d-2}(-t)}{\sum_{i=1}^{m} \mathbb{I}\left(\sqrt{n}|\hat{\alpha}_i|/\sqrt{\hat{\rho}\hat{\sigma}_i^2} > t\right)}.$$

Here $\mathbb{I}$ denotes the indicator function. To show the asymptotic FDR control as $m, n \to \infty$, we take a Bayesian perspective by assuming that the parameters $\alpha_i$'s are independently generated from a common distribution. The key result is summarized in the following theorem and technical details can be found in the supplementary note.

**Theorem 1** *Let $\rho$ be the $(1,1)$th element of $\{\mathbb{E}(\mathbf{z}_s \mathbf{z}_s^\top)\}^{-1}$. Suppose the following conditions are satisfied:*

*(i) $\mathbf{z}_s$'s are i.i.d.; $u_s$ and $c_{sa}, a = 1, ..., d$, are sub-Gaussian; $\sigma_{min}\{\mathbb{E}(\mathbf{z}_s \mathbf{z}_s^\top)\} > C$, where $\sigma_{min}(\mathbf{A})$ represents the minimum eigenvalue of a matrix $\mathbf{A}$.*

*(ii) $\sigma_i$'s are i.i.d. and $\mathbb{P}(C_1 < \sigma_i < C_2) = 1$.*

*(iii) $\varepsilon_{is}/\sigma_i \sim^{i.i.d.} \mathcal{E} =^d N(0,1)$ for $i = 1, ..., m$ and $s = 1, ..., n$.*

*(iv) $\alpha_i$'s are i.i.d.*

*(v) $\mathbf{z}_s, \sigma_i, \varepsilon_{is}/\sigma_i$, and $\alpha_i$ for $i = 1, ..., m$ and $s = 1, ..., n$ are mutually independent.*

*(vi) Denote by $f_n(\cdot; a)$ the density function of $\sqrt{n}\alpha_i + \sqrt{a}\varepsilon_{is}$ for any $a > 0$. For large enough $n$, the density $f_n(\cdot; \rho)$ has a unique mode at 0, i.e., $\arg\max_{x \in \mathbb{R}} f_n(x; \rho) = 0$; for any $\epsilon > 0$, there exists a $\delta > 0$ such that $\min_n \inf_{|x| > \epsilon} |f_n(x; \rho) - f_n(0; \rho)| > \delta$.*

*(vii) The Fourier transform $k(u) = \int_{-\infty}^{\infty} e^{-\iota u y} K(y) dy$ is absolutely integrable, where $\iota = \sqrt{-1}$ is the imaginary unit.*

*(viii) $h = o(1)$ and $1/(mh^2) = o(1)$.*

*(ix) $m = o(e^{Cn})$.*

*(x) Let $S_{\infty,n}(t) = \mathbb{P}(|\mathcal{E} + \sqrt{n}\alpha_i/\sqrt{\rho\sigma_i^2}| > t)$. There exists $t_0$ such that for large enough $n$, $2F_{n-d-2}(-t_0)/S_{\infty,n}(t_0) \leq q$.*

*Let*

$$FDR_{m,n}(t) = \mathbb{E}\left\{ \frac{\sum_{i:\alpha_i=0} \mathbb{I}\left(|\sqrt{n}\hat{\alpha}_i|/\sqrt{\hat{\rho}\hat{\sigma}_i^2} > t\right)}{1 \vee \sum_{i=1}^{m} \mathbb{I}\left(|\sqrt{n}\hat{\alpha}_i|/\sqrt{\hat{\rho}\hat{\sigma}_i^2} > t\right)} \right\}.$$

*Under the above conditions, we have*

$$\limsup_{m \to \infty, n \to \infty} FDR_{m,n}(t^*) \leq q.$$

Conditions (i)–(v) help prove the consistency of the variance estimators and the mode of the regression coefficients. By assuming that the errors follow the normal distributions (Condition (iii)), we can integrate all the relevant covariate information in a single parameter $\hat{\rho}$, which facilitates the establishment of the consistency of the kernel density estimation and hence the estimator of mode. In the simulation studies, we also investigated the scenario of non-normal distribution. We use an example to illustrate Condition (vi). In particular, we assume that $\sqrt{n}\alpha_i$ follows a discrete distribution with $\mathbb{P}(\sqrt{n}\alpha_i = a_{n,l}) = \pi_l$ for $l = 0, 1$, where $a_{n,0} = 0$, $a_{n,1} \neq 0$, $\pi_l > 0$, and $\pi_0 + \pi_1 = 1$. To reflect the sparsity, $\pi_0$ is set to be 0.8. We choose $a_{n,1} = 2$ and 5 representing weak and strong signals, respectively. We consider two cases for the error variance: (i) $\sigma_i = 1$; (ii) $\sigma_i \sim \mathrm{IG}(a, b)$, i.e., $\sigma_i$ follows the inverse-gamma distribution with the shape parameter $a$ and scale parameter $b$. As seen from Fig. 5, when the signal strength is weak, the mode of $\sqrt{n}\alpha_i + \sqrt{\rho}\varepsilon_{is}$ slightly deviates from 0 as the blue curve in the left panel indicates. For strong signals, the mode is exactly equal to zero. As shown in [43], Condition (vii) is fulfilled by many commonly used kernels such as the Gaussian kernel and the uniform kernel on $[-1, 1]$. Condition (ix) allows the number of taxa to be exponentially larger than the sample size. Condition (x) ensures the existence of a cut-off value to control the FDR at level $q$. A similar assumption was imposed in Theorem 4 of [44].

**Detailed setups for numerical studies**

The differential taxa were randomly drawn from the entire set. In particular, let $H_i = 0$ if the $i$th taxon is differentially abundant and $H_i = 1$ otherwise. The underlying truth $H_i$ was generated from

$$H_i \sim^{\text{i.i.d.}} \text{Bernoulli}(\gamma).$$

We simulated two levels of signal density (i.e., percentage of differential taxa) $\gamma = 5\%$, 20%, roughly corresponding to sparse and dense signals. We assumed that the baseline absolute abundance $X_{is}^{(0)}$ follows

$$\log\left(X_{is}^{(0)}\right) \sim^{\text{i.i.d.}} N\left(\beta_i^{(0)}, \sigma_i^2\right),$$

and correspondingly the absolute abundance $X_{is}$ were draw based on

$$\log(X_{is}) \sim^{\text{i.i.d.}} N\left(\beta_i^{(0)} + u_s\alpha_i + \mathbf{c}_s^\top \boldsymbol{\beta}_i^{-(0)}, \sigma_i^2\right),$$

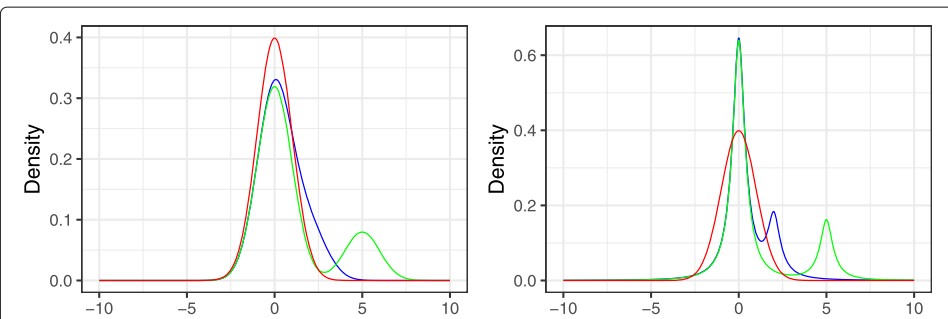

**Fig. 5** Density of $\sqrt{n}\alpha_i + \varepsilon_{is}$. The panels on the left and right correspond to $\sigma_i = 1$ and $\sigma_i \sim \mathrm{IG}(2, 1)$ respectively, where IG denotes the inverse-gamma distribution. The red curve is the density of the standard normal distribution. The blue and green curves are the densities of $\sqrt{n}\alpha_i + \varepsilon_{is}$ with $\mathbb{P}(\sqrt{n}\alpha_i = 0) = 0.8$ and $\mathbb{P}(\sqrt{n}\alpha_i = 2) = 0.2$, and $\mathbb{P}(\sqrt{n}\alpha_i = 0) = 0.8$ and $\mathbb{P}(\sqrt{n}\alpha_i = 5) = 0.2$, respectively

where $\boldsymbol{\beta}_i^{-(0)}$ represents the coefficients of the confounders, $i = 1, \ldots, m$. Let

$$\pi_{is}^{(0)} = \frac{X_{is}^{(0)}}{\sum_{j=1}^{m} X_{js}^{(0)}} \quad \text{and} \quad \pi_{is} = \frac{X_{is}}{\sum_{j=1}^{m} X_{js}}.$$

The observed OTUs data were simulated by

$$(Y_{1s}, \ldots, Y_{ms}) \sim^{\text{i.i.d.}} \text{Multinomial}(N_s, \pi_{1s}, \ldots, \pi_{ms}).$$

To create a power curve, we included six effect sizes labeled as $\{1, 2, \ldots, 6\}$ in the figures. We made the effect sizes have the same signs for differential taxa (i.e., the differential taxa have the same direction of change), creating a relatively strong compositional effect. Since low-abundance taxa have much less statistical power, we up-weighted their effects so that the power will not be dominated by those abundant ones. Specifically, for a randomly drawn differential taxon $i$, we set

$$\alpha_i = \begin{cases} \log(2\mu)\mathbb{I}\left(\bar{\pi}_i^{(0)} > 0.005\right) + \log\left\{2\mu\left(0.005/\bar{\pi}_i^{(0)}\right)^{1/3}\right\}\mathbb{I}\left(\bar{\pi}_i^{(0)} \leq 0.005\right) \text{ for } n = 50, \\ \log(\mu)\mathbb{I}\left(\bar{\pi}_i^{(0)} > 0.005\right) + \log\left\{\mu\left(0.005/\bar{\pi}_i^{(0)}\right)^{1/3}\right\}\mathbb{I}\left(\bar{\pi}_i^{(0)} \leq 0.005\right) \text{ for } n = 200, \end{cases}$$

where $\mu$ is equally spaced on $[1.05, 2]$ and $\bar{\pi}_i^{(0)} = \sum_{s=1}^{n} \pi_{is}^{(0)}/n$. We considered three cases for the covariate and confounders:

C0.  *A binary covariate.* $u_s \sim^{\text{i.i.d.}}$ Bernoulli(1/2) and no confounder.
C1.  *A continuous covariate.* $u_s \sim^{\text{i.i.d.}} N(0, 1)$ and no confounder.
C2.  *A binary covariate of interest and two confounders.*
    $u_s \sim$ Bernoulli($\{1 + \exp(-0.5c_{s1} - 0.5c_{s2})\}^{-1}$) independently, where $c_{s1}$ and $c_{s2}$ are confounders (i.e., $\mathbf{c}_s = (c_{s1}, c_{s2})^\top$). In the above, $c_{s1}$ is specified to independently follow the Rademacher distribution and $c_{s2} \sim^{\text{i.i.d.}} N(0, 1)$. The corresponding coefficients of the confounders $\boldsymbol{\beta}_i^{-(0)} = (\beta_i^{(1)}, \beta_i^{(2)})^\top$, $i = 1, \ldots, m$, were independently generated from a 2-dimensional normal distribution with mean $(1, 2)^\top$ and variance $\mathbf{I}_2$, where $\mathbf{I}_2$ denotes the 2 by 2 identity matrix.

The parameters $\beta_i^{(0)}$, $\sigma_i^2$, and $N_s$ were generated based on the estimation for a real dataset (COMBO) from the study of the gut microbiota in a general population [45], which consists of 98 samples and 6674 taxa. We only used its 500 most abundant taxa. Since $\beta_i^{(0)}$ and $\sigma_i^2$ were not directly estimable using the relative abundance data, we estimated $\beta_i^{(0)} - \beta_j^{(0)}$ and $\sigma_i^2 + \sigma_j^2$ based on the pairwise log ratios, forced some $\beta_i^{(0)}$'s to be zeros to obtain the estimators of $\beta_1^{(0)}, \ldots, \beta_m^{(0)}$, and derived $\sigma_i^2$ from the values of $\{\sigma_i^2 + \sigma_j^2\}_{i,j}$. We assume that the library size for each sample follows the negative binomial distribution

$$N_s \sim^{\text{i.i.d.}} \text{NB}(7645, 5.3),$$

where the mean and dispersion parameters were estimated based on the combo data. The resulting sparsity (percent of zeros) of the count matrix is around 65–75%.

In addition to the basic setting (S0, log normal abundance distribution), we designed seven other settings to study the robustness of the proposed method. Specifically, on top of S0 and C0 (a binary covariate), we studied

S1.  *zero-inflated absolute abundances.* The microbiome data contains excessive zeros and many zeros in the microbiome data can be explained by insufficient sampling [46] since majority of the taxa are of low-abundance. However, it is also possible that

zeros are due to physical absence of the taxa [47]. To study the effect of zero inflation on differential abundance analysis, we randomly forced 30% of the absolute abundance data to be 0.

S2. *Correlated absolute abundances.* Existing differential abundance analysis methods assume independence among taxa. However, in practice, taxa are interconnected forming networks [48]. It is interesting to see if the methods compared are robust to the correlations among the taxa. In this setting, we simulated block-correlation structure by dividing the 500 taxa into 25 equal-sized blocks. Within each block, we further divided the block into 2 by 2 sub-blocks and simulated equal positive correlations (0.5) within each sub-block and equal negative correlations ($-0.5$) between the two sub-blocks. This mimics the scenario that there are mutualistic relationships within the group and competitive relationships between groups.

S3. *Gamma abundance distribution.* Although the log normal distribution has been widely used for modeling species abundance data, other models such as gamma distribution are also possible [49]. We thus did additional simulation studies using the gamma distribution. Let $X_{is}^{(0)} \sim^{\text{i.i.d.}} \text{Gamma}(\eta_i^{(0)}, 1)$ and $X_{is} \sim^{\text{i.i.d.}} \text{Gamma}(\eta_i^{(0)} \exp(u_s \alpha_i + \mathbf{c}_s^\top \boldsymbol{\beta}_i^{-(0)}), 1)$. Similarly, we estimated $\eta_i^{(0)}$ from the COMBO data, where we first estimated the baseline proportion $\pi_i^{(0)}$ based on the Dirichlet-multinomial distribution using the R function `dirmult` and set the over-dispersion parameter $\theta^{(0)}$ to be 0.003, then let $\eta_i^{(0)} = \pi_i^{(0)}(1/\theta^{(0)} - 1)$.

S4. *Smaller m.* In microbiome data, each taxon can be assigned a taxonomic lineage and taxa abundances can be aggregated at different taxonomic ranks. Differential abundance analysis at higher ranks such as family and genus is also routinely performed. At the higher ranks, the number of taxa is much smaller. We thus studied a small number of taxa ($m = 50$) to see if the proposed method is robust to a small $m$. In this setting, we randomly chose 50 elements from $\boldsymbol{\beta}^{(0)} = (\beta_1^{(0)}, ..., \beta_{500}^{(0)})^\top$ and $\boldsymbol{\sigma}^2 = (\sigma_1^2, ..., \sigma_{500}^2)^\top$ in each simulation run. We set $N_s \sim \text{NB}(1500, 5.3)$.

S5. *Smaller n.* In pilot microbiome studies, the sample sizes are usually small. It is interesting to study the performance of the methods at a much smaller sample size. We studied $n = 20, 30$, and used the same effect size as $n = 50$.

S6. *10-fold difference in library size.* When the microbiome samples are not fully randomized in sequencing, it is likely that samples of the two groups end up in two separate sequencing runs leading to very different library sizes for the two groups. Since the presence/absence of a taxon strongly depends on the library size, the differential library size will confound the two-sample comparison, especially for those rare taxa [25]. To create differential library sizes, we generated the library size from $N_s \sim \text{NB}(5000, 5.3)$ and $N_s \sim \text{NB}(50000, 5.3)$ for the two groups, respectively.

S7. *Negative bionomial abundance distribution.* DESeq2 and edgeR assume negative binomial distribution for the counts, thus we included one more simulation setting, where we generated the counts from the negative binomial distribution. Let $X_{is}^{(0)} \sim^{\text{i.i.d.}} \text{NB}(\exp(7645\kappa_i^{(0)}), \theta_i^{(0)})$ and $Y_{is} \sim^{\text{i.i.d.}} \text{NB}(\exp(N_s\kappa_i^{(0)} + u_s\alpha_i + \mathbf{c}_s^\top \boldsymbol{\beta}_i^{-(0)}), \theta_i^{(0)})$. Similarly, we estimated the $\kappa_i^{(0)}$ (regression coefficient for the library size with respect to the log of the count of $i$th taxon) and $\theta_i^{(0)}$ (dispersion parameter) from the COMBO data using the R function `glm.nb`.

S8.   *Mixed-effect model.* We considered two scenarios: *Pre-treatment and post-treatment comparison* (S8.1) and *Replicate sampling* (S8.2). Under S8.1, for $n = 50$ (or 200), we simulated 25 (or 100) subjects and each has paired pre-treatment and post-treatment samples. The aim is to detect taxa affected by treatment. Under S8.2, each subject has multiple measurements. For $n = 50$ (or 200), we generated 25 (or 50) subjects with each having 2 (or 4) replicates. Specifically, we let

$$\log(X_{is}) \sim \mathbf{r}_s^\top \boldsymbol{\gamma}_i + N(\beta_i^{(0)} + u_s\alpha_i + \mathbf{c}_s^\top \boldsymbol{\beta}_i^{-(0)}, \sigma_i^2),$$

where $\mathbf{r}_s$ has one element equal to 1 and all the others equal to 0 indicating the subject ID of sample $s$. Each element of $\boldsymbol{\gamma}_i$ follows $N(0, \tau_i^2)$ independently, where we let $\tau_i^2 = a_i\sigma_i^2$ with $a_i \sim \text{Unif}([0, 1])$.

## Supplementary Information

---

**Additional file 1:** Supplementary notes. **Table S1** lists some robust normalization methods [11–14]. **Lemmas S1 – S4** present intermediate results for proving Theorem 1. The proof depends on some useful results from [50–53].

**Additional file 2:** Supplementary figures. **Fig. S1** and **S2** compare the proposed method LinDA with different zero-handling approaches under settings S6C0 and S0C0. **Fig. S3** depicts the results of LinDA, CLR-OLS and MaAsLin2 with different normalization approaches under setting S0C0. **Fig. S4–S10** and **S12–S14** show the results of settings S0C1, S0C2, S1C0, S2C0, S4C0, S5C0, S6C0, S7C0, S8.1C0, and S8.2C0, respectively. The comparison between disabling and enabling zero treatment of the ANCOM-BC method is depicted in **Fig. S11** under setting S6C0. **Fig. S15** shows the results of setting S0C0 with stronger compositional effects. **Fig. S16–S19** show the effect size plots and volcano plots for the four datasets (CDI, IBD, RA, and SMOKE) respectively. **Fig. S20–S30** present the full result of all methods under different simulation settings.

**Additional file 3:** Review history.

---

## Acknowledgements
Not applicable.

## Peer review information

## Review history
The review history is available as Additional file 3.

## Authors' contributions
J.C. and X.Z. conceived, designed and supervised the work together. X.Z. developed the method. X.Z., K.H., and H.Z. performed the theoretical analysis. H.Z. and J.C. performed the evaluation and developed the software. J.C., H.Z., X.Z., and K.H wrote and revised the manuscript together. The authors read and approved the final manuscript.

## Funding
This work was supported by National Institute of Health R01GM144351 (Chen & Zhang), National Science Foundation DMS-1830392, DMS2113359, DMS1811747 (Zhang & Zhou) and National Science Foundation DMS2113360 and Mayo Clinic Center for Individualized Medicine (Chen).

## Availability of data and materials
LinDA is implemented as the `linda` function in the CRAN R package `MicrobiomeStat` (https://CRAN.R-project.org/package=MicrobiomeStat). The `LinDA` package is also available at GitHub (https://github.com/zhouhj1994/LinDA) [54]. The entire codes and data for generating the presented results are available at the repository Zenodo (https://doi.org/10.5281/zenodo.6326019) and at GitHub (https://github.com/zhouhj1994/LinDA-manuscript-result) under MIT License [55]. The CDI, IBD, RA, SMOKE and COMBO datasets are from [29–32, 45], respectively.

## Declarations

### Ethics approval and consent to participate
Not applicable.

### Consent for publication
Not applicable.

**Competing interests**
The authors declare that they have no competing interests.

**Author details**
[1]Shanghai University of Finance and Economics, Shanghai 200437, China. [2]Texas A&M University, College Station 77843, USA. [3]Renmin University of China, Beijing 100872, China. [4]Mayo Clinic, Rochester, USA.

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

## 