## [**Additional file 3** Review history. · Genome Biology]

Review History

First round of review

Reviewer 1

Were you able to assess all statistics in the manuscript, including the appropriateness of statistical tests used? Yes. The statistical procedures seem appropriate.

Were you able to directly test the methods? No.

Comments to author:

This manuscript presents a new method, LinDA for differential abundance testing in taxonomic profiling studies with compositional data. Differential abundance analysis is one of the key approaches in current taxonomic profiling studies based on amplicon or metagenomic sequencing. A vast array of methods have been proposed and benchmarked for this task during the past decade. Most of the current studies include compositional data, where we have access only to proportions, and not absolute abundances of the different taxonomic groups. The statistical challenges related to compositionality are well recognized. The authors propose a relatively simple but flexible approach that addresses these challenges in compositional microbiome data analysis. The method is a linear regression model applied on CLR transformed data, with an additional compositional bias correction for which the authors are providing an asymptotic derivation. The method is shown to have a convenient asymptotic FDR control property and extendability to mixed-effect models with correlated microbiome data. The method performance is benchmarked against currently popular alternative methods using simulations and real data of varying dimensionalities, and these comparisons support earlier reports, and are also as such useful for the field; in the presented experiments LinDA is systematically one of the best performing methods, and its findings seem to overlap relatively well with the alternatives. An advantage of a close competitor, ANCOM-BC, is substantially improved scalability in LinDA.

This is interesting and well written work that appears technically sound with appropriate, well described, and properly validated methods. The proposed method provides a plausible alternative for currently popular methods of microbiome DA testing. The work is original, and appropriate references are provided. I have some suggestions and reservations, however.

Major comments

1) The detected DA taxa from LinDA seem to overlap relatively well with the alternatives. It would be informative to see examples of cases where LinDA results differ from the other methods (failure to identify a DA detected by other methods; or positively detected DA taxa that is not found by other methods. Could we have some illustrations, e.g. jitterplots, in the supplementary material showing how the taxon abundance are distributed in the two groups in such cases. This could give more intuition on where the method succeeds, and what might be its weaknesses.

2) Many DA studies, including those cited by the authors (e.g. DESeq2, edgeR) have assumed negative binomial distribution for the counts instead of Gamma distribution. Can the authors comment on why negative binomial has not been used/included in the comparisons, and how this might change the results?

3) Some more discussion on the weak points of LinDA could be added; to inform the reader in which situations the method might fail.

4) This method is potentially a valuable addition to the toolkit of microbiome data scientists, if the benefits will be proven to be robust in subsequent independent benchmarkings. If the authors have confidence on the performance of the method and its potential to become a widely used technique for DA testing, this would be greatly facilitated by providing a ready-made general-purpose implementation, for instance in the form of an R package; such packages are already available for all the main methods that were included in the comparisons. The lack of implementation will severely limit the practical value of the proposed method among the relevant research communities. The implementation would also facilitate further, independent comparisons by other researchers and thus replication of the findings and conclusions. In the present submission, only the source code is provided and a general purpose implementation is missing.

Minor comments

Line 38: could you explain both numerator and denominator; now N_s is explained but it is not intuitively clear what the more complex dividing term does. Is this the maximum read count across all samples, or something else? Was this approach proposed in Brill (2020) or designed by the authors of the current manuscript? Brill is cited for

reported shortcomings but not cited for this solution, it would be useful to know how the authors ended up with this new imputation approach.

Section 4.6 line 48-49; "most of the zeroes" -> I think "many zeroes" would be more appropriate as this depends on the data/environment, and other considerations, and if we only detect zeroes, it is problematic to make statements that most zeroes are false as a general rule.

Code for the experiments is provided in:

<https://github.com/zhouhj1994/LinDA-manuscript-result> The value of this code would be improved by adding an open source license, see e.g. DOI: 10.1371/journal.pcbi.1002598 and the long-term preservation of the exact version used in this manuscript would be best guaranteed by adding DOI to this code repository, for instance through Zenodo (as is now generally recommended), see e.g. <https://guides.github.com/activities/citable-code/>

Table 1: the presentation could be revised; I found the use of different numbers of stars and circles to denote different degrees of FDR control and inflation difficult to grasp and compare across different methods intuitively, and the Table is understandable only when reading the caption carefully. It is not clear from the table what is the corresponding numerical performance.

Figs 1-2: "False Discovery Proportion" -> Should this be "False Discovery Rate" or "False Negative Rate"?

Figure 4: consider using UpSet plot as an alternative to Venn diagram

<https://www.ncbi.nlm.nih.gov/pmc/articles/PMC4720993/>

Reviewer 2

Were you able to assess all statistics in the manuscript, including the appropriateness of statistical tests used? Yes. Reviewed.

Were you able to directly test the methods? Yes.

Comments to author:

Huijuan Zhou, Kejun He, Jun Chen, and Xianyang Zhang in their paper entitled 'LinDA: Linear Models for Differential Abundance Analysis of Microbiome Compositional Data' propose a linear regression framework for differential abundance analysis that purportedly fills a methodological gap for the compositionality of 16S microbiome. Additionally, the github

repository is straight forward, but further documentation, eg. vignette or submission to CRAN or Bioconductor would be more appropriate.

The work is overall interesting and well written, but I have several suggestions/criticisms that I hope will help improving the manuscript.

Major Comments:

- A philosophical topic: It's been of consider discussion whether the compositionality is relevant, e.g. <https://link.springer.com/article/10.1186/s12864-018-5160-5> or if preferential sequencing removes any compositionality as observed in the count data <https://www.ncbi.nlm.nih.gov/pmc/articles/PMC6531881/>
- There have been a number of benchmarking papers on the topic of differential abundance and method comparison. Notably, <https://link.springer.com/article/10.1186/s40168-016-0208-8> that highlight the role sparsity has played an impact on the field. Other articles have focused on types of normalization combined with inference method, <https://link.springer.com/article/10.1186/s40168-017-0237-y> . Could the authors please comment on these (and other) settings with respect to the benchmarking comparison and / or strategy?
- It is also unclear why the authors left several methods following an initial one out of the majority of comparisons. Could the authors include all appropriate methods when applicable?
- This is perhaps out of scope, but I have noticed many researchers are using <https://huttenhower.sph.harvard.edu/maaslin/> . Would authors be able to include this tool in their assessment?

No minor items.

Cover Letter

We thank the AE and two reviewers for the insightful comments and valuable suggestions. We have made a careful revision following the comments and suggestions from both reviewers. We mention a few major changes as follows:

1. We licensed the source code for the manuscript and added the DOI. We integrated `LinDA` into our CRAN package `MicrobiomeStat` as the ‘`linda`’ function.
2. We evaluated a new method `MaAsLin2`, and included one more simulation setting which generates the counts from a negative binomial distribution.
3. We added more discussions on the weakness of our method.

Response to AE

1. Referee 2 recommends additional comparisons and also has comments on the method accounts for sparsity and normalization.

Ans: We compared `LinDA` to `MaAsLin2` and found that `LinDA` outperforms `MaAsLin2` with different normalization approaches. For more details, see Discussion section and Supplementary Figure S16 of the revision.

2. Referee 1 suggests describing examples where `LinDA` identifies something that other methods do not, thereby highlighting strengths and weaknesses of the different approaches.

Ans: We now indicated the differential taxa detected solely by `LinDA` on the effect size plots for the real examples and provided some interpretation.

3. Referee 2 also recommends implementing an R package.

Ans: We have uploaded our package LinDA to the github repository (<https://github.com/zhouhj1994/LinDA>). We also integrated it into our CRAN MicrobiomeStat package (<https://CRAN.R-project.org/package=MicrobiomeStat>).

4. We are considering your manuscript as a 'Method' article. This is our format for publishing articles that describe a methodological innovation that is a significant advance over published methods and likely to be of broad utility, but that do not provide significant biological insights. When revising the manuscript, please ensure the manuscript conforms to our style for Methods articles (see <https://genomebiology.biomedcentral.com/submission-guidelines/preparing-your-manuscript/method>); specifically, the abstract should be under 100 words. Please note that if we decide to publish your manuscript we will require that the source code is made publicly available under an open source license compliant with Open Source Initiative, with the license clearly stated in the manuscript. The source code should be deposited in a public repository, such as for instance github, with the accession links included in the manuscript. We also ask that the version of source code used in the manuscript is deposited in a DOI-assigning repository, such as zenodo, with the link also included. All this information should be listed in a separate Availability of Data and Materials section of the manuscript.

Ans: Thanks for the information. The manuscript has been written in the format of a "Method" article. Scripts and complete results for the manuscript are available on GitHub (<https://github.com/zhouhj1994/LinDA-manuscript-result>) with open source license (MIT License) and DOI:10.5281/zenodo.5635982.

Response to Reviewer 1

Major comments

1. The detected DA taxa from LinDA seem to overlap relatively well with the alternatives. It would be informative to see examples of cases where LinDA results differ from the other methods (failure to identify a DA detected by other methods; or positively detected DA taxa that is not found by other methods. Could we have some illustrations, e.g. jitterplots, in the

supplementary material showing how the taxon abundance are distributed in the two groups in such cases. This could give more intuition on where the method succeeds, and what might be its weaknesses.

Ans: Thanks for this nice suggestion. Since we aim for detecting taxa which are differential on the absolute abundance level, the plots based on relative abundances (i.e., proportions) may not necessarily reflect the true difference on the absolute abundances, esp. when the compositional effect is strong. Moreover, when there are covariates, the jitterplots/boxplots may not be able to adjust covariate effects. We thus believe that the effect size plots before and after the bias correction (Supplementary Figures S17–S19) could give more insights into our method. In the effect size plots, we now indicated the taxa which are detected by LinDA only (in red). We can see that, for the taxa solely detected by LinDA, the effect sizes tend to be underestimated without bias correction and bias correction improves the power in these cases. On the contrary, for the taxa missed by LinDA but detected by ANCOM-BC and one more method (in blue), the effect sizes tend to be overestimated without bias correction.

2. Many DA studies, including those cited by the authors (e.g. DESeq2, edgeR) have assumed negative binomial distribution for the counts instead of Gamma distribution. Can the authors comment on why negative binomial has not been used/included in the comparisons, and how this might change the results?

Ans: Thanks for your comment. We have included one more simulation setting, which generates the counts from a negative binomial distribution. The results show that DESeq2 and edgeR continue to suffer from large FDR inflation due to the strong compositional effects in the data. For more information, see Discussion section and Supplementary Figure S15 of the revision.

3. Some more discussion on the weak points of LinDA could be added; to inform the reader in which situations the method might fail.

Ans: We have added some discussion about the limitations of LinDA so that the user is well informed when selecting the best differential abundance analysis tool for their dataset. Please see the texts highlighted in red in the discussion. The major limitations of LinDA are (1) some FDR inflation when the number of taxa is small, (2) the log linear assumption may be violated and (3) the sparsity assumption may not hold. We thus do not recommend LinDA to datasets with

small taxa numbers (e.g., phylum-level analysis). For those significant taxa detected by LinDA, we also suggest the users to perform model checking using the linear model-based diagnostic tools provided by R.

4. This method is potentially a valuable addition to the toolkit of microbiome data scientists, if the benefits will be proven to be robust in subsequent independent benchmarkings. If the authors have confidence on the performance of the method and its potential to become a widely used technique for DA testing, this would be greatly facilitated by providing a ready-made general-purpose implementation, for instance in the form of an R package; such packages are already available for all the main methods that were included in the comparisons. The lack of implementation will severely limit the practical value of the proposed method among the relevant research communities. The implementation would also facilitate further, independent comparisons by other researchers and thus replication of the findings and conclusions. In the present submission, only the source code is provided and a general purpose implementation is missing.

Ans: Thanks for your suggestion. We have uploaded our package LinDA to a github repository (<https://github.com/zhouhj1994/LinDA>). We also integrated it into our CRAN MicrobiomeStat package (<https://CRAN.R-project.org/package=MicrobiomeStat>).

Minor comments

1. Line 38: could you explain both numerator and denominator; now N_s is explained but it is not intuitively clear what the more complex dividing term does. Is this the maximum read count across all samples, or something else? Was this approach proposed in Brill (2020) or designed by the authors of the current manuscript? Brill is cited for reported shortcomings but not cited for this solution, it would be useful to know how the authors ended up with this new imputation approach.

Ans: The denominator is the maximum read count across samples where the specific taxon is not present. It is a new approach designed by the authors to reduce the false positive rate when the sequencing depth is correlated with the variable of interest. Brill (2020) uses a rarefaction approach, which tends to be less powerful as we found. In this new approach, we treat zeros as

left-censored missing data and try to impute them based on the information available. Suppose we only know the sequencing depth and want to guess the true values behind the zeros. A natural strategy is to make the imputed value proportional to the sequencing depth with the highest depth sample receiving a fractional count close to 1 (in our approach, we simply set it as 1). We have explained our procedure more intuitively in the text.

2. Section 4.6 line 48-49; "most of the zeroes" → I think "many zeroes" would be more appropriate as this depends on the data/environment, and other considerations, and if we only detect zeroes, it is problematic to make statements that most zeroes are false as a general rule.

Ans: Fixed. Thanks.

3. Code for the experiments is provided in: <https://github.com/zhouhj1994/LinDA-manuscript-result> The value of this code would be improved by adding an open source license, see e.g. DOI: 10.1371/journal.pcbi.1002598 and the long-term preservation of the exact version used in this manuscript would be best guaranteed by adding DOI to this code repository, for instance through Zenodo (as is now generally recommended), see e.g. <https://guides.github.com/activities/citable-code/>

Ans: Thanks for your suggestion. Scripts and outputs for the manuscript are available on GitHub (<https://github.com/zhouhj1994/LinDA-manuscript-result>) with open source license (MIT License) and DOI:10.5281/zenodo.5635982.

4. Table 1: the presentation could be revised; I found the use of different numbers of stars and circles to denote different degrees of FDR control and inflation difficult to grasp and compare across different methods intuitively, and the Table is understandable only when reading the caption carefully. It is not clear from the table what is the corresponding numerical performance.

Ans: We have split the table into two sub-tables to compare the empirical FDR and true positive rate separately. Table 1A compares the empirical FDR. Three ★ represents that the FDR is controlled; two , one and zero ★ represent slight, large and severe FDR inflation, respectively. Table 1B compares the true positive rate. More ★ represents higher power compared to the other methods.

5. Figs 1-2: "False Discovery Proportion" → Should this be "False Discovery Rate" or "False

Negative Rate”?

Ans: The “False Discovery Proportion” has been changed to “Empirical False Discovery Rate”.

6. Figure 4: consider using UpSet plot as an alternative to Venn diagram <https://www.ncbi.nlm.nih.gov/pmc/articles/PMC4720993/>

Ans: Thanks for this suggestion. We have substituted the Venn diagram with UpSet plot. See Figure 4 of the revision.

Response to Reviewer 2

Major comments

1. The github repository is straight forward, but further documentation, eg. vignette or submission to CRAN or Bioconductor would be more appropriate.

Ans: Thanks for the comment. We have now integrated the procedure into our CRAN package MicrobiomeStat (<https://CRAN.R-project.org/package=MicrobiomeStat>).

2. A philosophical topic: It’s been of considerable discussion whether the compositionality is relevant, e.g. <https://link.springer.com/article/10.1186/s12864-018-5160-5> or if preferential sequencing removes any compositionality as observed in the count data <https://www.ncbi.nlm.nih.gov/pmc/articles/PMC6531881/>

Ans: Thanks for the insightful comment. We now elaborated more on this topic in the discussion and introduction. Compositionality is more relevant when analyzing individual microbial features since the major interest to biomedical investigators is to find those truly differential features (“driver”) instead of those driven by the compositional effect (“passenger”). However, for community-wide analysis such as distance-based analysis, addressing compositionality may not be necessary in order to control the type I error, since compositional effect is only relevant under the alternative hypotheses. Considering compositionality in the community-wide analysis has also been found to have small effects on the statistical power (Thorsen et al., 2016; Weiss et al., 2017). Additionally, in microbiome-based predictive models, the relative abundance and/or

their ratios could already be informative features for prediction and addressing compositionality may not necessarily increase the prediction accuracy significantly. Therefore, whether to address compositionality depends on the specific problems. The experimental method presented in <https://www.ncbi.nlm.nih.gov/pmc/articles/PMC6531881/> makes the measured composition (relative abundances) more close to that in the sampling site, but it could not be used to obtain the absolute abundances per se. To obtain the absolute abundance data, one needs to measure the total microbial load in the sample. There are several experimental techniques such as qPCR, spike-in and flow cytometry can achieve this goal. However, due to their limitations as described in Morton et al. (2019), none of these procedures have been widely used in practice, making addressing compositional effects through computational approach the dominant approach.

3. There have been a number of benchmarking papers on the topic of differential abundance and method comparison. Notably, <https://link.springer.com/article/10.1186/s40168-016-0208-8> that highlight the role sparsity has played an impact on the field. Other articles have focused on types of normalization combined with inference method, <https://link.springer.com/article/10.1186/s40168-017-0237-y> . Could the authors please comment on these (and other) settings with respect to the benchmarking comparison and / or strategy?

Ans: The benchmarking paper (<https://link.springer.com/article/10.1186/s40168-016-0208-8>) used a real data-based simulation strategy, where the samples were randomly selected from a real dataset and differential signals were added to a subset of taxa. Since no parametric model was assumed in generating the counts, the simulated data could preserve the major characteristics of the real data such as the high sparsity. However, one drawback of this approach is the way it simulates differential signals, where the counts of the taxa are multiplied by a fold change so zero counts remain zeros. In reality, a large number of zeros are due to under-detection and as the abundance increases, they could become non-zeros. In contrast, the benchmarking paper (<https://link.springer.com/article/10.1186/s40168-017-0237-y>) used parametric models including multinomial, Dirichlet-multinomial and Gamma-Poisson models to simulate the counts. In our benchmarking study, we also used the parametric approach: a log normal model was used to simulate the absolute abundances and multinomial model was then used to mimick the sequencing process. Since the parametric model may not fully capture the data characteristics, we conducted more robustness studies including using an alternative parametric model on the absolute abundance (Gamma), simulating correlations among taxa and increasing the sparsity

level by adding more zeros. These robustness studies were used to complement the shortcomings of parametric models.

4. It is also unclear why the authors left several methods following an initial one out of the majority of comparisons. Could the authors include all appropriate methods when applicable?

Ans: Thanks for your comment. We found that DESeq2, edgeR and metagenomeSeq-2 had severe FDR inflation under most settings. To make the figures more clear, we did not include them in the main comparison and focused on the comparison between LinDA, ANCOM-BC, ALDEx2, metagenomeSeq, and Wilcoxon, which have far better false positive control than DESeq2, edgeR and metagenomeSeq-2. Full results of all methods are available at <https://github.com/zhouhj1994/LinDA-manuscript-result>.

5. This is perhaps out of scope, but I have noticed many researchers are using <https://huttenhower.sph.harvard.edu/maaslin/>. Would authors be able to include this tool in their assessment?

Ans: Thanks for your suggestion. MaAsLin2 is also based on log linear model. However, it does not address the compositionality specifically. In the default implementation, the total sum scaling normalization is used. We now compared LinDA to MaAsLin2 and found that LinDA outperforms MaAsLin2 with different normalization approaches. For more details, see Discussion section and Supplementary Figure S16 of the revision.

References

Morton, J. T., Marotz, C., Washburne, A., Silverman, J., Zaramela, L. S., Edlund, A., Zengler, K., and Knight, R. (2019). Establishing microbial composition measurement standards with reference frames. *Nature communications* **10**, 2719.

Thorsen, J., Brejnrod, A., Mortensen, M., Rasmussen, M. A., Stokholm, J., Al-Soud, W. A., Sørensen, S., Bisgaard, H., and Waage, J. (2016). Large-scale benchmarking reveals false discoveries and count transformation sensitivity in 16S rRNA gene amplicon data analysis methods used in microbiome studies. *Microbiome* **4**, 62.

Weiss, S., Xu, Z. Z., Peddada, S., Amir, A., Bittinger, K., Gonzalez, A., Lozupone, C., Zan-

eveld, J. R., Vázquez-Baeza, Y., Birmingham, A., et al. (2017). Normalization and microbial differential abundance strategies depend upon data characteristics. *Microbiome* **5**, 27.

Second round of review

Reviewer 1

I would like to thank the authors for comprehensively addressing the earlier comments. I still have the following remarks.

Major comments

1) PDF page 18, line 4: this cites the github site for various important supplementary materials. This has critical shortcomings: 1) These results are not explained (no figure or table captions, or detailed README), this makes it challenging to read these results, clicking through dozens of individual result files with no explanation; 2) There are no guarantees on long-term preservation on github site, although adding DOI solves this issue. I strongly propose that the main results from github repository will be moved to the formal supplementary material in order to guarantee unambiguous documentation and long-term availability.

2) MaAsLin2 is so commonly used that it would be appropriate to include in the main comparisons, not just supplements.

3) LinDA Github package (<https://github.com/zhouhj1994/LinDA>) or MicrobiomeStat CRAN version with the linda function (<https://cran.r-project.org/web/packages/MicrobiomeStat/index.html>) are not supporting commonly used taxonomic abundance data representation formats in R, i.e. phyloseq or (Tree)SummarizedExperiment. The former is a more widely adopted, and the latter is a newer format with an increasing user base. Supporting one or both of these formats would make the methods more readily accessible for much broader audience. Many of the comparison methods are already implementing such support. I do not think that this is critical for the current manuscript but essential to consider otherwise for overall impact and utility of this work.

4) Table 1A/B: not clear how to mild/medium/severe have been determined. Is there a systematic criterion? The figure caption does not explain the row acronyms, it is not possible to understand the current version of this table in a self-contained way. Consider other notation than multiple stars? Is it possible to represent these quantitative results as a barplot or lineplot?

Minor comments

pdf p 5; l. 22-37: this seems a long and technical explanation for introductory section on an already commonly known issue i.e. compositional data; key references would be sufficient.

pdf p 6; l. 4: not sure if it is necessary to write we do not claim that our model is “correct” as this applies to all or most of scientific data analysis

Acronyms like "S6C0" are confusing in the text; is it possible to provide a more descriptive name?

Table 2: is it possible to bolden the best results, for better readability?

Reviewer 2

Thank you to the authors for their comments. There are still a few outstanding remarks to make.

Major:

- The simulation framework is not convincing as representative for microbiome data given the past decade's work on simulations and method development. I believe this tool to be useful for the community, but a few of the statements are very strong and recommend toning them accordingly unless going back to other simulation frameworks and including comparisons of methods to those.

Minor:

- The authors have called throughout metagenomeSeq and metagenomeSeq2 to refer to fitFeatureModel and fitZig respectively. These are actually the opposite per documentation / prior literature. Swapping terminology will be important for consistency.

- Phrasing in places can sometimes be confusing

Response to reviewers' comments

We thank the reviewers for the constructive and helpful comments, which gives us another chance to improve the clarity of our manuscript. Below are our point-to-point responses to the reviewers' comments.

Reviewer #1

Major comments

1) PDF page 18, line 4: this cites the github site for various important supplementary materials. This has critical shortcomings: 1) These results are not explained (no figure or table captions, or detailed README), this makes it challenging to read these results, clicking through dozens of individual result files with no explanation; 2) There are no guarantees on long-term preservation on github site, although adding DOI solves this issue. I strongly propose that the main results from github repository will be moved to the formal supplementary material in order to guarantee unambiguous documentation and long-term availability.

Response: Thanks for this thoughtful comment. We now included the full results in the supplementary file under the section "S3 Full comparisons of numerical studies".

2) MaAsLin2 is so commonly used that it would be appropriate to include in the main comparisons, not just supplements.

Response: We now added MaAsLin2 in the main comparisons. See the updated figures and texts.

3) LinDA Github package (<https://github.com/zhouhj1994/LinDA>) or MicrobiomeStat CRAN version with the linda function (<https://cran.r-project.org/web/packages/MicrobiomeStat/index.html>) are not supporting commonly used taxonomic abundance data representation formats in R, i.e. phyloseq or (Tree)SummarizedExperiment. The former is a more widely adopted, and the latter is a newer format with an increasing user base. Supporting one or both of these formats would make the methods more readily accessible for much broader audience. Many of the comparison methods are already implementing such support. I do not think that this is critical for the current manuscript but essential to consider otherwise for overall impact and utility of this work.

Response: Thanks for this constructive comment! We now added support for phyloseq object.

4) Table 1A/B: not clear how to mild/medium/severe have been determined. Is there a systematic criterion? The figure caption does not explain the row acronyms, it is not possible to understand the current version of this table in a self-contained way. Consider other notation than multiple stars? Is it possible to represent these quantitative results as a barplot or lineplot?

Response: Since the definition of "mild/medium/severe" is very subjective and qualitative, we now remove this table to avoid potential confusion and inaccurate representation. We believe that the summary in the discussion could give the reader a similar overview of the pros and cons of various methods.

Minor comments

pdf p 5; l. 22-37: this seems a long and technical explanation for introductory section on an already commonly known issue i.e. compositional data; key references would be sufficient.

Response: Thanks for the suggestion. We now deleted the lengthy description and key references were added.

pdf p 6; l. 4: not sure if it is necessary to write we do not claim that our model is “correct” as this applies to all or most of

Response: We now deleted this sentence.

Acronyms like "S6C0" are confusing in the text; is it possible to provide a more descriptive name?

Response: We now added a short description following those setting notations.

Table 2: is it possible to bolden the best results, for better readability?

Response: Done. Thanks.

Reviewer #2

Major:

- The simulation framework is not convincing as representative for microbiome data given the past decade's work on simulations and method development. I believe this tool to be useful for the community, but a few of the statements are very strong and recommend toning them accordingly unless going back to other simulation frameworks and including comparisons of methods to those.

Response: Thanks for this helpful and constructive comments. We now toned down some statements and acknowledged the limitation of the simulation framework in the discussion. Please see the added discussion in the manuscript, which is also pasted below:

“Although the presented simulation settings could give basic insights into the performance of various methods, such model-based simulations might not be able to capture the full characteristics of the real microbiome data. It is very likely that the performance of the compared methods will change using a different simulation framework. Moreover, our simulation strategy purposely creates strong compositional effects, where all differential taxa show the same direction of change. Such setting is used to test the limit of the various methods in addressing the compositional effects. However, in real data, the compositional effects may not be always strong, and the FDR inflation of many methods could be very moderate. Therefore, a future benchmarking study, which uses real data-based simulation strategy and investigates all biologically plausible differential settings, is much needed to have a comprehensive and objective evaluation of existing differential abundance analysis methods.”

Minor:

- The authors have called throughout metagenomeSeq and metagenomeSeq2 to refer to fitFeatureModel and fitZig respectively. These are actually the opposite per documentation / prior literature. Swapping terminology will be important for consistency.

Response: Done. Thanks for pointing this out.

- Phrasing in places can sometimes be confusing

Response: We have proofread the manuscript several times and corrected some potential confusing statements.